



# Competing influences of the ocean, atmosphere and solid earth on transient Miocene Antarctic ice sheet variability

Lennert B. Stap[1], Constantijn J. Berends[1], Meike D.W. Scherrenberg[1], Roderik S.W. van de Wal[1,2], and Edward G.W. Gasson[3]

[1]Institute for Marine and Atmospheric research Utrecht, Utrecht University, 3584 CC Utrecht, the Netherlands
[2]Faculty of Geosciences, Department of Physical Geography, Utrecht University, Utrecht, the Netherlands
[3]College of Life and Environmental Sciences, University of Exeter, Exeter, United Kingdom

**Correspondence:** L.B. Stap (L.B.Stap@uu.nl)

**Abstract.** Benthic $\delta^{18}$O levels vary strongly during the warmer-than-modern early- and mid-Miocene (23 to 14 Myr ago), suggesting a dynamic Antarctic ice sheet (AIS). So far, however, realistic simulations of the Miocene AIS have been limited to equilibrium states under different $CO_2$ levels and orbital settings. Earlier transient simulations lacked ice-sheet-atmosphere interactions, and used a present-day rather than Miocene Antarctic bedrock topography. Here, we quantify the effect of ice-

sheet-atmosphere interactions, running IMAU-ICE using climate forcing from Miocene simulations by the general circulation model GENESIS. Utilising a recently developed matrix interpolation method enables us to interpolate the climate forcing based on $CO_2$ levels (between 280 and 840 ppm) as well as ice sheet configurations (between no ice and a large ice sheet). We further-more implement recent reconstructions of Miocene Antarctic bedrock topography. We find that the positive albedo-temperature feedback, partly compensated by the negative ice-volume-precipitation feedback, increases hysteresis in the relation between

$CO_2$ and ice volume (V). Together, these ice-sheet-atmosphere interactions decrease the amplitude of AIS variability caused by 40-kyr forcing $CO_2$ cycles by 21% in transient simulations. Thereby, they also diminish the contribution of AIS variability to benthic $\delta^{18}$O fluctuations. Furthermore, we show that under equal atmospheric and oceanic forcing, the amplitude of 40-kyr transient AIS variability becomes 10% smaller during the early- and mid-Miocene, due to the evolving bedrock topography. Lastly, we quantify the influence of ice shelf formation around the Antarctic margins, by comparing simulations with Last

Glacial Maximum (LGM) basal melt conditions, to ones in which ice shelf growth is prevented. Ice shelf formation increases hysteresis in the $CO_2$-V relation, and amplifies 40-kyr AIS variability by 19% using LGM basal melt rates, and by 5% in our reference setting.

## 1 Introduction

The dynamics of the Antarctic ice sheet (AIS) during the early- and mid-Miocene (23 to 14 Myr ago) are a topical issue in

(paleo)climate science. This issue is of great interest, because the climatic conditions of this period, such as the atmospheric $CO_2$ concentration (Kürschner et al., 2008; Foster et al., 2012; Badger et al., 2013; Greenop et al., 2014; Super et al., 2018; Steinthorsdottir et al., 2021b), ranged from those of the recent past to those predicted for the future (~200 to >800 ppm, see Steinthorsdottir et al. (2021a), for a review). Benthic $\delta^{18}$O records show sizeable orbital-timescale fluctuations during this time,



an indication of potential vigorous Antarctic ice volume variability (e.g. Zachos et al., 2008; Liebrand et al., 2011, 2017; Hol-
bourn et al., 2013; Levy et al., 2019). The $\delta^{18}O$ signal is, however, also shaped by deep-sea temperature changes. How much of
the signal is committed by each component, and thus how the AIS evolved during the Miocene, is currently still under debate.
On the one hand, geological evidence supports strong AIS dynamism (Pekar and DeConto, 2006; Shevenell et al., 2008), with
ice periodically advancing and retreating e.g. over Wilkes Land (Sangiorgi et al., 2018) and the Ross Sea sector (Hauptvogel
and Passchier, 2012; Levy et al., 2016; Pérez et al., 2021). Some studies, on the other hand, advocate a more stable small
AIS, based on ice-proximal sea-surface temperature records from dinoflagellate cyst assemblages (Bijl et al., 2018) and TEX$_{86}$
data (Hartman et al., 2018). In addition, a recent analysis of clumped isotope and Mg/Ca records has found high bottom-water
temperatures, which combined with the benthic $\delta^{18}O$ record paradoxically suggests a Miocene AIS volume persistently larger
than at present day (Modestou et al., 2020). Finally, using climate modelling and data comparisons, it has been suggested that
the AIS varied mostly in spatial extent, rather than volume, which imprinted on Miocene $\delta^{18}O$ variability through a strong
effect on deep-water temperatures (Bradshaw et al., 2021).

Ice-sheet models can be used to understand the physical processes driving Miocene AIS variability. Broadly stated, earlier
modelling efforts have followed two approaches to this task. One approach is to use ice-sheet models bidirectionally coupled
to climate models, with as much physical realism, in terms of processes included and resolution, as is computationally feasible
for long-term simulations. Gasson et al. (2016) adopted this approach, using an isotope-enabled setup of a 3D ice-sheet model
coupled to a regional climate model nested in a general circulation model (GCM). They showed the importance of including
ice-sheet-climate feedbacks, ice-calving parameterisations, and changes in the ice sheet's oxygen isotopic composition, for
simulating strong AIS-induced benthic $\delta^{18}O$ changes. Halberstadt et al. (2021) used the same model set-up to determine the
influence of CO$_2$ changes and uplift of the Transantarctic Mountains on glacial variability during the Miocene. These studies
were so far limited to steady-state simulations, in which the climate forcing is kept constant until the ice sheet has equilibrated.
Transient ice sheet behaviour can be very different, as was e.g. demonstrated for the North American and Eurasian ice sheets
during the Late Pleistocene (Abe-Ouchi et al., 2013) . The second approach, which is the one taken in this study, is therefore to
focus on transient AIS changes, applying gradually changing climatic conditions. Earlier studies employed 1D ice-sheet mod-
els in combination with parameterised temperatures (De Boer et al., 2010), or relatively simple climate models (Langebroek
et al., 2009, 2010; Stap et al., 2017). Recently, Stap et al. (2019) showed that orbital-timescale CO$_2$ changes cause smaller tran-
sient AIS variability than equilibrium simulations suggest, using the 3D thermodynamical Parallel Ice Sheet Model (PISM)
forced by Miocene climates obtained from GCM (COSMOS) simulations (Stärz et al., 2017). Conceptual considerations in-
dicate that the disequilibrium between ice volume and CO$_2$ also leads to out-of-phase changes of these quantities (Stap et al.,
2020). However, the COSMOS climate simulations were performed using only one ice sheet configuration. Hence, the effect
of surface whitening (darkening) by ice sheet advance (retreat) on temperature, the so-called albedo-temperature feedback, was
not included. Neither was the ice-volume-precipitation feedback, which causes precipitation on the ice sheet to decrease with
increasing ice volume. Furthermore, the effect of mass balance changes due to surface height changes (surface-height-mass-
balance feedback) could only be crudely captured using a scalar lapse-rate correction for temperature, which is known to be a





simplification (Fortuin and Oerlemans, 1990). Moreover, a present-day Antarctic bedrock topography was used, while during
the Miocene the topography was different, which also significantly affects the surface mass balance and ice flow (Stap et al.,
2016; Colleoni et al., 2018; Paxman et al., 2020).

In this study, we examine the effect of ice-sheet-atmosphere feedbacks, as well as bedrock topography, on transient Miocene
AIS variability. We run the 3D thermodynamical ice-sheet model IMAU-ICE (De Boer et al., 2013; Berends et al., 2018), that
is based on the commonly-used shallow ice and shallow shelf approximations and therefore comparable to PISM. Climate
forcing is obtained from Miocene simulations by the GCM GENESIS, in which different $CO_2$ levels and ice sheet configura-
tions ranging from zero ice to a substantial ice volume are applied (Burls et al., 2021). Utilising a recently developed matrix
interpolation routine (Berends et al., 2018), this set-up enables us to interpolate the climate forcing based on land ice changes
in addition to $CO_2$ changes. In this manner, the albedo-temperature and ice-volume-precipitation feedbacks are represented,
and because surface height changes are incorporated in the forcing, the surface-height-mass-balance feedback is captured more
realistically as well. Performing several experiments that include both equilibrium and transient simulations, we test the in-
fluence of these feedbacks under varying $CO_2$ levels. Furthermore, we quantify the effect on transient Miocene AIS volume
evolution of using bedrock topographies (Hochmuth et al., 2020a; Paxman et al., 2019) pertaining to the early- (23-24 Myr ago)
and mid-Miocene (14 Myr ago), as well as the Eocene (34 Myr ago) and present day. Lastly, we investigate where and under
which conditions ice shelves form, and what their influence on the grounded ice volume is. This is achieved by performing and
comparing simulations, in which relatively mild, and severe basal melt rates underneath the ice shelves are applied.

## 2 Models and methods

For this study, we employ the ice-sheet model IMAU-ICE (v1.1.1-MIO) (De Boer et al., 2013; Berends et al., 2018) to simulate
the Antarctic ice sheet, using atmospheric input data obtained from Miocene climate simulations by the GCM GENESIS (Burls
et al., 2021).

### 2.1 Ice-sheet model

IMAU-ICE is a threedimensional thermodynamical ice-sheet model, which uses a superposition of the shallow ice approxi-
mation (SIA) and shallow shelf approximation (SSA) to simulate the dynamics of grounded and floating ice. In the interior of
the ice sheet, the ice flow is dominated by the SIA, while towards the fast-flowing ice streams on the fringes the SSA gains
importance, becoming the governing component on the ice shelf (Winkelmann et al., 2011). We use a 40 x 40-km resolution
grid covering the Antarctic continent. This resolution ensures the feasibility of performing a large numbers of simulations,
while still capturing the Antarctic ice flow in the amount of detail we deem appropriate for our purpose. For further settings,
please see Appendix A



### 2.1.1 Basal mass balance

Basal melt underneath the ice shelves (M) is calculated using a combination of the parameterisation of Pollard and DeConto (2009) and a linear relation to ocean temperature change (Beckmann and Goosse, 2003; Martin et al., 2011):

$$M = (z_{deep} - 1)[(1 - z_{expo})S_m + z_{expo}M_{expo}] - (z_{deep}M_{deep}),$$ (1)

with

$$S_m = \rho_w c_p \gamma_T \frac{T_w - T_{freeze}}{H_{fus}\rho_i}.$$ (2)

Here, the weighing factors $z_{deep} = \frac{h_w - 1800\,\text{m}}{200\,\text{m}}$ and $z_{expo} = \frac{\alpha_{sub} - 80°}{30°}e^{-0.4*r_{open}}$ depend on the water depth ($h_w$), the widest subtended angle to the open ocean ($\alpha_{sub}$) and the shortest linear distance to the open ocean ($r_{open}$). The melt rate for the exposed shelves ($M_{expo}$) varies linearly between 3 m/yr for an applied $CO_2$ concentration of 280 ppm to 6 m/yr for $CO_2$ concentrations of 400 ppm and higher. The melt rate for the deep-water shelves ($M_{deep}$) similarly varies between 5 and 10 m/yr. Melt rate $S_m$ is dependent on the water density ($\rho_w$), specific heat ($c_p$), thermal exchange velocity ($\gamma_T$), water tempera-

ture ($T_{water}$), pressure-adjusted freezing point ($T_{freeze}$), latent heat of fusion ($H_{fus}$) and shelf ice density ($\rho_i$). Similar to melt rates $M_{expo}$ and $M_{deep}$, $T_w$ varies linearly as a function of $CO_2$ between -1.7°C (280 ppm) and 2°C ($\geq$ 400 ppm).

Currently, calving is neglected. In combination with the heavily parameterised sub-shelf melt calculation, this makes the mass balance of the ice shelves arguably a weak point of our modelling effort. Therefore, in a separate sensitivity experiment

(Table 1), we consider an extreme case using Last Glacial Maximum (LGM)-like melt parameter settings: $M_{expo} = 0\,\text{m/yr}$, $M_{deep} = 2\,\text{m/yr}$, and $T_w = -5°\text{C}$ (De Boer et al., 2013) . In another experiment, ice shelves are effectively inhibited from growing by applying a constant melt rate of $400\,\text{m/yr}$ similar to the ABUM experiment of the ice-sheet model intercomparison project ABUMIP (Sun et al., 2020).

### 2.1.2 Surface mass balance

The surface mass balance (SMB) calculation uses precipitation ($P_{applied}$) and 2-m air temperature ($T_{applied}$), obtained from Miocene climate simulations by GENESIS (Sect. 2.2) utilising a matrix interpolation routine (Appendix B). Monthly SMB is the sum of accumulation and refreezing, minus ablation (Berends et al., 2018). Refreezing is dependent on the available liquid water content, and the superimposed water content (Huybrechts and de Wolde, 1999; Janssens and Huybrechts, 2000). Accumulation ($Acc$) is calculated as:

$$Acc = P_{applied} * f_{snow},$$ (3)

with $f_{snow}$ the temperature-dependent snow fraction (Ohmura, 1999). Ablation ($Abl$) follows from an insolation-temperature melt parameterisation (Bintanja et al., 2002):

$$Abl = c_1(T_{applied} - 273.15\,\text{K}) + c_2(Q_{TOA} * (1 - \alpha)) + c_3,$$ (4)



where $\alpha$ is the internally-calculated surface albedo, and the parameters are set to $c_1 = 0.788 \, \mathrm{m \, yr^{-1} \, K^{-1}}$, $c_2 = 0.004 \, \mathrm{m^3 \, J^{-1}}$,
and $c_3 = -0.50 \, \mathrm{m \, yr^{-1}}$. Incoming solar radiation at the top of the atmosphere $Q_{TOA}$ is taken from Laskar et al. (2004), and
kept constant at the present-day values, so as to isolate climate variability caused by $CO_2$ changes. This surface mass balance
parameterisation has recently been shown to yield realistic results over the present-day Greenland ice sheet (Fettweis et al.,
2020).

## 2.2 Climate model

GENESIS (version 3.0) (Thompson and Pollard, 1997; DeConto et al., 2012) has a spectral resolution of T31 ($\sim 3.75° \times 3.75°$).
The atmospheric model is coupled to $2° \times 2°$ surface models including a non-dynamical 50-m slab diffusive mixed-layer ocean
with dynamical sea ice, and dynamic vegetation (BIOME4). The applied paleogeography comes from Herold et al. (2008),
with dedicated modifications to Antarctica. Different simulations were carried out as described in Burls et al. (2021); a subset
is used in this study. The simulations used here have a distinct $CO_2$ level (280 ppm, 840 ppm), Antarctic ice sheet configuration
(volumes of 0 km$^3$ (no ice) and $18.975 \times 10^6$ km$^3$ (full ice sheet)), and orbital settings (eccentricity = 0.05, obliquity = 22.5°,
longitude of precession = 90° (cold Antarctic summer), ecc. = 0, obl. = 23.5°, longitude of precession = N/A (medium, modern
day settings), ecc. = 0.05, obl. = 24.5°, longitude of precession = 270° (warm Antarctic summer)).

For our reference simulations, we use the 2-m air temperature, precipitation, and surface height fields of simulations 1fumebi
(cold) and 3nomebi (warm), bilinearly interpolated to the ice-sheet model grid (Fig. S1). Simulation 1fumebi has 280 ppm $CO_2$
(1 x pre-industrial (PI) level), a full ice sheet, and medium Antarctic summers, while simulation 3nomebi has 840 ppm $CO_2$
(3 x PI), no ice sheet, and medium Antarctic summers. The suffix 'bi' in the names of the GENESIS simulations indicates that
dynamic vegetation model BIOME4 was used, which is of no direct concern for the current study.

## 2.3 Simulations

We perform an experiment using our default settings (REF), as well as 12 sensitivity experiments with settings altered as
indicated in Table 1. Each experiment consists of 15 simulations, 11 of which are steady-state, and 4 of which are transient
simulations (Table 2).

The steady-state simulations are run for 150 kyr with a constant forcing $CO_2$ level; enough time for the ice sheet to equili-
brate with the forcing climate. Mind that the $CO_2$ level, rather than the climate itself, is kept fixed, which means the climate
can still change due to land ice changes. First, six of these steady-state simulations are run. In our reference case, the initial
conditions are obtained from recent reconstructions of the Antarctic bathymetry (Hochmuth et al., 2020a) and bedrock topog-
raphy (Paxman et al., 2019) pertaining to 23 to 24 million years (Myr) ago (dataset: Hochmuth et al., 2020b). No ice is present
at the start of these runs. The $CO_2$ levels are set to different levels bracketing the 280 to 840 ppm range: 280, 392, 504, 616,
728, and 840 ppm. These simulations constitute the lower, ascending branch of the hysteresis curve in the equilibrium $CO_2$-ice





volume relation (blue lines in figures, e.g. Fig. 2b). Thereafter, the remaining five steady-state simulations are continued from results of the 280-ppm IMAU-ICE run, with $CO_2$ levels of 392, 504, 616, 728, and 840 ppm. These simulations constitute the upper, descending branch of the hysteresis curve in the equilibrium $CO_2$-ice volume relation (red lines in figures, e.g. Fig. 2b).


Next, we perform the transient simulations. Three of these are continued from the 840-ppm IMAU-ICE run. The $CO_2$ is first linearly reduced from 840 to 280 ppm, and then increased back to 840 ppm, over the course of 100, 400 kyr, and five consecutive times over 40 kyr (200 kyr total). The final transient simulation is continued from the 280-ppm IMAU-ICE run. In this simulation, $CO_2$ is first gradually increased from 280 to 840 ppm, and then reduced back to 280 ppm five consecutive

times over 40 kyr. This is essentially the normal transient 40-kyr run mirrored.

## 3 Results

### 3.1 Reference simulations

As a minimal check for our model and parameter settings, we first perform steady-state simulations using ERA40 present-day

(PD; 1957-2002) precipitation and temperature forcing (Uppala et al., 2005). Depending on whether the run is started from Bedmachine PD ice and topography (Morlighem et al., 2020) or without ice and an isostatically rebounded topography, we simulate a total ice volume above floatation of 56.6 meter sea level equivalent (m.s.l.e.; $23.1 \times 10^{15}$ m$^3$) or 55.0 m.s.l.e. ($22.4 \times 10^{15}$ m$^3$) respectively. This is very close to the 55.0 m.s.l.e. obtained from remapping the Bedmachine data to our grid. Like other SIA/SSA-based ice-sheet models (c.f. De Boer et al., 2015), IMAU-ICE generally simulates slightly thinner ice in the interior

and thicker ice along the margins (Fig. S2). The grounding line is simulated quite well, although slightly more advanced in the Filchner-Ronne, and the Amery embayments. Due to lack of calving, the simulated ice shelves are extended much farther than the observations. However, since the increase in back pressure due to the extra shelf area is very small (Reese et al., 2018a), this is unlikely to affect the evolution of the grounded ice. In this study, we will therefore focus on grounded ice.

In the reference equilibrium simulations using Miocene forcing (experiment REF), the AIS consists only of disconnected glaciers in high mountainous regions in case of a 840-ppm $CO_2$ level. It grows into a full East AIS (EAIS) and small West AIS (WAIS) in case of 280-ppm forcing (Fig. 1). In the latter case, the WAIS grounds on Marie Byrd Land and the Peninsula, while not growing into the Ross and Filchner-Ronne embayments. The reference $CO_2$-equilibrium ice volume (above floatation) ($V_{eq}$) relation decreases monotonically and shows considerable hysteresis (Fig. 2b). Depending on the timescale of the

imposed forcing variability, the transient simulations yield peak ice volumes of 45.48 m.s.l.e. (400 kyr) and 37.81 m.s.l.e. (100 kyr). In agreement with Stap et al. (2020), this peak is only reached after the $CO_2$ minimum (Fig. 2a), and relatively earlier, i.e. at a lower $CO_2$ level, in the 400-kyr simulation than in the shorter 100-kyr simulation. In case of 40-kyr forcing cycles, the equilibrium cycle is attained very quickly and does not depend on the initial conditions (Fig. 2c,d). In the equilibrium cycle, the ice volume varies between 3.6 m.s.l.e. at 728 ppm to 30.7 m.s.l.e. at 532 ppm, a 27.1 m.s.l.e. difference. Since the





maximum ice volume is reached longer after the minimum $CO_2$ level (252 ppm difference) than the minimum ice volume after the maximum $CO_2$ level (112 ppm difference), the growth phase lasts longer than the decay phase. That also means that ice sheet decay is on average 40% faster than growth, but both rates, in particular the growth rate, are certainly not constant. In the transient simulations, the AIS initially builds up slowly in the high mountainous regions at ∼0.1 m.s.l.e./kyr, speeding up to a maximum of 3 m.s.l.e./kyr when the surface mass balance (SMB) becomes positive on a large part of the continent as separate

ice domes start to merge. When the transient ice volume is in between the branches of the $CO_2$-$V_{eq}$ relation, the rate of ice volume change is relatively small (<1 m.s.l.e./kyr).

In earlier research using IMAU-ICE, different values for tuning parameter $c_3$ in Eq. 4 have been used (e.g. De Boer et al., 2013, 2014; Berends et al., 2018). Our setting $C_3 = 0.50 \, \mathrm{m \, yr^{-1}}$ leads to satisfactory PD results, but using $C_3 = 0.25 \, \mathrm{m \, yr^{-1}}$ as

De Boer et al. (2014) did, a similar PD ice volume of 55.5 m.s.l.e. is obtained (not shown). Using this setting (experiment SMB; Table 1) increases surface melt, with as a result onset of AIS growth and decay at lower $CO_2$ levels (Fig. 3a,b). We choose our REF setting of $C_3 = 0.50 \, \mathrm{m \, yr^{-1}}$, such that variability in our simulated $CO_2$ domain is maximised. A different tuning might lead to quantitatively different $CO_2$ levels, but not qualitatively. Furthermore, in the REF simulations we ignore the effect of orbital forcing variability completely. Performing experiment INSOL (Table 1), however, we make an assessment of the

indirect effect of insolation changes (through temperature) on Miocene AIS variability. We now interpolate between the results of GENESIS simulations 1fucobi (cold Antarctic summer) and 3nowabi (warm Antarctic summer). Mind that the influence of insolation on SMB (through the second term on the r.h.s. of Eq. 4) remains constant at PD level. The indirect insolation effect changes in concert with $CO_2$; we therefore recalculate the weight given to $CO_2$ changes compared to ice-extent changes in our interpolation method (Eq. B5) as $\epsilon_{CO2} = 0.72$ in this case. In these simulations, the maximum equilibrium ice volume (at 280

ppm) is larger: 54.2 m.s.l.e. compared to 48.4 m.s.l.e. in case REF. This volume increase is established mostly in regions with low bedrock topography, through a thicker WAIS in Marie Byrd Land and Ellsworth Land, and EAIS in MacRobertson Land and Coats Land (not shown). Furthermore, hysteresis in the $CO_2$-$V_{eq}$ relation is diminished, in consonance with Langebroek et al. (2009), and transient ice volume variability is amplified (Fig. 3c,d). In this study, we perform idealised simulations, focussing on influences of the ocean, atmosphere and solid earth on transient Miocene AIS variability. When aiming to make

realistic simulations, however, the INSOL simulations indicate that the matrix interpolation routine should be extended to include orbital forcing.

## 3.2 Influence of interaction between ice sheets and atmosphere

Using a matrix interpolation routine enables us to interpolate the climate forcing for our ice-sheet model based on both $CO_2$ and ice volume. Here, we make a comparison to results of our model set-up using a simpler glacial index routine, as e.g. used in

Stap et al. (2019), in which the interpolation is solely based on the $CO_2$ level. In experiment FEEDB (Table 1), we implement $w_{tot} = \frac{CO_2 - CO_{2,cold}}{CO_{2,warm} - CO_{2,cold}}$ instead of Eqs. B7 (for temperature) and B11 (for precipitation) to interpolate between the cold and warm climates. Because the positive albedo-temperature feedback is not invoked in this case, hysteresis in the $CO_2$-$V_{eq}$ relation is smaller (Fig. 4b). The temperature declines more quickly towards lower $CO_2$ values, causing the equilibrium simulations to



yield larger ice volumes. At 280 ppm, the difference in ice thickness with the REF simulation is again mainly manifested in
regions with low bedrock topography (not shown). This also results in larger transient ice volume variability (Fig. 4a,b). Forced
by 40-kyr $CO_2$ cycles, the amplitude of variability is 34.1 m.s.l.e. in the equilibrium cycle, 26% larger than in the REF case.
Stated the other way around, the ice-sheet-atmosphere interactions decrease the amplitude of AIS variability by 21%.

As mentioned before, the difference between the REF and FEEDB results is mainly governed by the albedo-temperature
feedback. However, in the REF case, precipitation decreases with increasing ice volume, via Eq. B10 combined with a drier
climate at lower $CO_2$. This constitutes a negative ice-volume-precipitation feedback. The effect of this feedback alone is shown
by a comparison to the results of experiment FEEDB onlyP (Table S1, Fig. 4c,d). In this experiment, the glacial index method
is only used for precipitation, and temperature is interpolated in the normal manner (Eq. B7). The hysteresis in the $CO_2$-$V_{eq}$
relation is now larger, and the amplitude of transient variability in the equilibrium cycle of the 40-kyr simulation is 19.3 m.s.l.e.,
29% smaller than in the REF case. In contrast, using the glacial index method only for temperature (experiment FEEDB onlyT),
the amplitude of 40-kyr transient variability becomes 6% larger (36.1 m.s.l.e.) than in the regular FEEDB case. In conclusion,
the albedo-temperature feedback is the strongest of these two ice sheet-atmosphere feedbacks in our experiments.

### 3.3    Influence of bedrock topography

Recent geological reconstructions have indicated that Antarctic bedrock topography has evolved substantially since the Eocene-
Oligocene boundary (34 Myr ago), mostly due to thermal subsidence and glacial erosion (Paxman et al., 2019). In the REF
simulations, we have used the bedrock topography reconstruction pertaining to the Oligocene-Miocene boundary (23-24 Myr
ago). In experiment TOPO 14Ma (Table 1), we instead use the one pertaining to the mid-Miocene (14 Myr ago). A larger part
of West Antartica (Marie Byrd Land and Palmer Land) has subsided below sea level in this reconstruction, which leads to
a smaller simulated WAIS at low $CO_2$ levels in this case (Fig. 5). The bedrock subsidence in Victoria Land and the Amery
embayment has a smaller effect on equilibrium ice volume. The total equilibrium ice volume above floatation is 44.6 m.s.l.e.
at 280 ppm, compared to 48.4 m.s.l.e. in the REF case (-8%). The amplitude of transient ice volume variability is affected as
well, as it decreases to 24.4 m.s.l.e (-10% w.r.t. REF) when 40-kyr forcing cycles are imposed (Fig. 6a,b).

The effect of long-term Antarctic landscape evolution on ice volume becomes more apparent when we compare simulations
using bedrock conditions of the Eocene-Oligocene boundary (34 Myr ago; TOPO 34Ma) and relaxed PD topography (TOPO
PD). In the latter case, less ice builds up in Victoria Land and Coats Land where the initial bedrock topography has subsided
below sea level, compared to the TOPO 14Ma case (Fig. S3a). The equilibrium ice volume at 280 ppm reduces to 42.8 m.s.l.e.
(-12% w.r.t. REF), and the amplitude of transient 40-kyr ice volume variability to 23.3 m.s.l.e. (-14% w.r.t. REF; Fig. 7a,b).
Conversely, it increases to 28.4 m.s.l.e. (+5% w.r.t. REF), and is centered around a generally larger ice volume, in experiment
TOPO 34Ma (Fig. 7a,b). This is also due to the fact that the EAIS is more stable, as shown by larger ice volumes already at
high $CO_2$ levels in the $CO_2$-$V_{eq}$ relation, particularly in the upper, descending branch. Hysteresis is indeed larger than in the
REF case, because the positive surface-height-mass-balance feedback is stronger due to the build-up of thicker ice. At 280
ppm, the equilibrium ice volume is 57.4 m.s.l.e. in this experiment (+19% w.r.t. REF). A much thicker ice sheet is built up



in Coats Land, the Amery embayment, and Marie Byrd Land (Fig. S3b). Still in this case, the ice sheet does not significantly advance into the Ross and Filchner-Ronne embayments. This is different when we use the older geological reconstructions of

Antarctic paleotopography at the Eocene-Oligocene boundary of Wilson et al. (2012). At 280 ppm, the ice then grounds in the Ross embayment (TOPO Wilson_mean, using the mean of the minimum and maximum reconstructions; Fig. (Fig. S3c)), or in both embayments (TOPO Wilson_max, using the maximum reconstruction; Fig. S3d). Due to the higher $CO_2$ levels at which deglaciation takes place, a relatively large ice volume persists throughout 40-kyr transient simulations (Fig. 7c,d). In contrast to the other experiments, the initial conditions of these transient simulations now affect the ice volume during the equilibrium

cycle. The ice sheet remains generally larger when the equilibrium 280-ppm simulation, rather than the 840-ppm simulation, is used as starting point (not shown).

### 3.4   Influence of ice shelf formation

We perform and compare two additional sensitivity experiments that are end-members with respect to basal melt rate settings, to assess the influence of ice shelf formation on AIS variability. In experiment BMB LGM (Table 1), we use Last Glacial Max-

imum (LGM)-like basal melt parameter settings in Eq. 1: $M_{expo} = 0 \, \text{m/yr}$, $M_{deep} = 2 \, \text{m/yr}$, and $T_w = -5°\text{C}$ (De Boer et al., 2013). The mild basal melt rates in this case create an environment in which ice shelves can grow relatively easily. At the other extreme, in experiment BMB no_shelves, ice shelves are effectively inhibited from growing by applying a constant melt rate of $400 \, \text{m/yr}$. The effect of ice shelf formation becomes notable at 504 ppm in the equilibrium simulations (lower, ascending branch) (Fig. 8a). Ice shelf form along the margins at Coates Land and Oates Land (Fig. 9a). While in Coates Land, this leads

to a modest thickening of the grounded ice, this happens to a much lesser degree in Oates Land. In the long 400-kyr transient simulation, these ice shelves form after around 110 kyr, at ~530 ppm (Fig. 8b). At 392 ppm in the equilibrium simulations, and after around 170 kyr at ~360 ppm in the 400-kyr transient simulation, ice shelves appear approximately simultaneously along almost the entire remainder of the (East and West) Antarctic margin (Fig. 9b). The effect on grounded ice is particularly notice-able in West-Antarctica, and Wilkes Land, George V Land and MacRobertson Land, where the ice sheet thickens by kilometers.

In the deglaciation phase these ice shelves remain influential until 728 ppm in the equilibrium simulations (upper, descending branch), and until around 360 kyr at ~728 ppm in the 400-kyr transient simulation. In the transient simulations, hysteresis in the $CO_2$-V relation, and the amplitude of variability are increased. In the 40-kyr transient simulations, the ice volume variability amounts to 30.7 m.s.l.e. in case LGM BMB and 25.7 m.s.l.e. in case LGM no_shelves, a 5 m.s.l.e. (19%) difference (Fig. 8c,d).

## 4   Discussion

The $CO_2$ levels at which East-Antarctic glaciation is simulated, are known to be dependent on the climate model used to construct the climatic forcing (Gasson et al., 2014). This could (partly) explain why EAIS inception occurs at higher $CO_2$ levels than in Stap et al. (2019). They used climate forcing from COSMOS which has a higher climate sensitivity (reported as 4.1 K per $CO_2$ doubling) than GENESIS (2.9 K). Note that in neither study greenhouse gasses other than $CO_2$ are varied, probably





biasing the $CO_2$ concentrations high. In contrast, our climate input comes from the same model as Halberstadt et al. (2021), who obtain relatively large EAIS volumes at $CO_2$ levels up to 1140 ppm. In absolute sense, this $CO_2$ difference can simply be caused by the value of the ablation factor that we use in our surface mass balance calculation ($C_3$ in Eq. 4), as demonstrated by our SMB sensitivity experiment (Fig. 3). Arguably the best way to further constrain this parameter in the future is by performing a comprehensive comparison of our mass balance calculation, and other methods and observations, as was done previously

for the Greenland ice sheet (Fettweis et al., 2020). Furthermore, they additionally use a regional climate model for downscaling and refrain from using bias corrections for the temperature and precipitation forcing. Nevertheless, these factors likely do not explain the narrower $CO_2$ range between inception of the East- and West-Antarctic ice sheet in our simulations compared to Halberstadt et al. (2021). At the root of this discrepancy can be the different schemes we use to calculate precipitation and ablation, which may be more sensitive to temperature changes. Differences in ice dynamics, e.g. in the treatment of ice flow

enhancement and basal sliding, might cause further disparity. Probably for the same reasons, we obtain larger transient AIS volume variability than Stap et al. (2019), when using a similar glacial index method (experiment FEEDB). Conducting an intermodel comparison of early AIS dynamics would be advisable for a more comprehensive investigation of the different results yielded by ice sheet models. We nonetheless corroborate earlier findings that disequilibrium between the AIS and the forcing climate causes smaller transient ice volume variability then steady-state simulations suggest (Stap et al., 2019), as well

as out-of-phase changes between these quantities (Stap et al., 2020). Here, we include ice-sheet-atmosphere feedbacks in our reference case. We have shown that these feedbacks tend to increase hysteresis in the $CO_2$-equilibrium ice volume relation, and decrease the amplitude of AIS variability under equal 40-kyr forcing $CO_2$ cycles by 21%. This implies that correspondingly, the contribution of AIS volume changes to benthic $\delta^{18}O$ fluctuations gets smaller, although the strength of the impact will depend on the isotopic composition of Antarctic snow as well (Gasson et al., 2016; Rohling et al., 2021).


Furthermore, we use recent reconstructions of Antarctic bedrock topography during the Miocene (Hochmuth et al., 2020a; Paxman et al., 2019), rather than the present day as in Stap et al. (2019), as a boundary condition in our AIS simulations. In agreement with Colleoni et al. (2018) and Paxman et al. (2020), we find that the WAIS becomes more vulnerable over the early- and mid-Miocene (Fig. 6). This is demonstrated by the lower equilibrium ice volumes at equal $CO_2$ levels in our experi-

ment TOPO 14Ma, which has a bedrock topography pertaining to the mid-Miocene (14 Myr ago), compared to REF, in which the topography is from the Miocene-Oligocene boundary (23-24 Myr ago). We also corroborate the finding of Paxman et al. (2020), that from 34 to 24 million years ago bedrock changes primarily concern the EAIS (Fig. 7), while from 24 Myr ago onwards mostly the WAIS is affected. Here, we have additionally examined the effect of the evolving Antarctic landscape on transient Miocene AIS variability, which is 10% smaller when subjected to 40-kyr forcing $CO_2$ cycles in TOPO 14Ma than in

REF, due to impeded WAIS growth. Hence, $CO_2$ minima must have dropped increasingly further over the course of the early- and mid-Miocene to obtain AIS variability with similar amplitude.

We simulate the inception of ice shelves only at relatively low $CO_2$ levels. Ice shelves first appear along the coasts of Oates Land and Coates Land when the $CO_2$ concentration drops below ~530 ppm, but the effect of ice shelves on grounded ice



volume only becomes significant below ∼360 ppm $CO_2$ when the EAIS - with a volume of almost 30 m.s.l.e. - is already well-established. This transition from a land-based to a marine ice sheet at $CO_2$ levels around 400 ppm is in general agreement with other model results (Halberstadt et al., 2021), and geological data inferences (Naish et al., 2009; Levy et al., 2016, 2019). Below ∼360 ppm $CO_2$, ice shelves fringe almost the entire continent in our simulations. This includes the Wilkes Subglacial Basin and the Ross Sea sector, where records from IODP Site U1356 and ANDRILL AND-2A show evidence of periodic ice

retreat and readvance (Hauptvogel and Passchier, 2012; Levy et al., 2016; Pérez et al., 2021; Sangiorgi et al., 2018). In the Ross and Filchner-Ronne embayments, however, growth of extended ice shelves never occurs in our Miocene simulations, not even when the basal melt rates are set to LGM values. On the one hand, it is certainly possible that our basal melt parameter-isation is not suited for the simulation of (large-scale) Antarctic ice shelf inception. On the other hand, the fact that the Ross and Filchner-Ronne ice shelf do grow when we impose steady present-day climatic forcing, points out that West-Antarctic

ice shelves have to be sustained by a sufficiently large ice flow coming from the WAIS. That means that in addition to the oceanic conditions, suitable atmospheric conditions are very important for ice shelf formation as well. Possibly, WAIS growth is therefore hampered by the air temperatures remaining too warm in our forcing climate simulations, because in the cold case (simulation 1fumebi) there is only a small WAIS present in the forcing climate simulation. In the future, the forcing steady-state climate simulations used in the matrix interpolation could be constructed by a coupled set-up of the same ice-sheet and climate

models that are deployed in the transient simulations. This would ensure congruence between the results of the equilibrium and transient simulations, so that the transient ice sheet will not grow beyond its largest size in the forcing equilibrium results.

When we increase $CO_2$ levels from cold conditions in our simulations, the ice shelves keep having an effect on grounded AIS ice volume until relatively high $CO_2$ levels of ∼728 ppm are reached. Only then does the AIS retreat into the Wilkes Subglacial

Basin, suggesting periodic AIS waxing and waning in this region has to be governed by substantial $CO_2$ fluctuations. Growth of ice shelves hence increases hysteresis in the $CO_2$-ice volume relation, in general agreement with the findings of Garbe et al. (2020). In line with this result, we find a different effect on final ice volume (above floatation) in steady-state PD simulations with a 400-m/yr melt rate, depending on whether the simulation is started with or without ice. Starting from an isostatically rebounded PD topography without ice, we obtain a volume of 39.5 m.s.l.e., 28% smaller than the PD run with normal basal

melt rates (Fig. S4a,c). When the simulation is started from a PD topography including grounded and floating ice, the final volume is 48.4 m.s.l.e., which is only 14% smaller than the normal PD run (Fig. S4b,d). Hence, in our model, the effect of ice shelves on grounded ice volume is smaller when existing ice shelves are melted away, than when ice shelves cannot form in the first place. Despite having a longer model runtime and different surface mass balance forcing, these simulations are similar to the ABUMIP ABUM experiment (Sun et al., 2020). Our simulated 8.2 m.s.l.e. loss of ice volume above floatation after ice

sheet removal under PD forcing conditions (starting with ice) is on the large side of the wide range of model responses (∼1 to 11 m.s.l.e. loss) reported there. Showing complete collapse of the WAIS and strong retreat in the Amery embayment, but limited ice loss in the Recovery, Wilkes and Aurora subglacial basins, our spatial response is most similar to that of ABUMIP models PISM1, PSU3D1, and BISICLES. However, our simulated response, using a Coulomb sliding law (q = 0.3), is weaker than that of ELMER/ICE and SICOPOLIS, that used a Weertman sliding law (m = 3), and that of a different version of IMAU-



ICE that used the Schoof flux boundary condition (Schoof, 2007) at the grounding line. A stronger response of grounded ice volume to ocean warming and ice shelf break-up might lead to earlier retreat, i.e. at a lower $CO_2$ level, of the AIS in e.g. the Wilkes Subglacial Basin, hence reducing hysteresis in the $CO_2$-ice volume relation.

Here, we have conducted idealised transient simulations, addressing the effects of the ocean, atmosphere and solid earth on Miocene AIS evolution. Moving forward towards realistic transient simulations of AIS evolution, our results of experiment INSOL indicate that orbital forcing variability will need to be accounted for in the model set-up (Fig. 3), for instance by integrating it in the matrix interpolation method (e.g. Ladant et al., 2014; Tan et al., 2018). A further improvement to be made is the implementation of a more advanced scheme to calculate the basal mass balance underneath the ice shelves; several alternatives have been developed in recent years (e.g. Reese et al., 2018b; Lazeroms et al., 2019; Pelle et al., 2019). This should be accompanied by ocean forcing supplied by sophisticated (3D) ocean models. As a final prospect, inclusion of a coupled sediment model will ultimately open up the possibility of quantifying the transient effect of changing bedrock topography on AIS dynamics over the Miocene (Pollard and DeConto, 2003, 2020).

## 5   Conclusions

Using the ice-sheet model IMAU-ICE, we have performed steady-state and transient simulations of the Antarctic ice sheet (AIS) during the early- and mid-Miocene (23 to 14 Myr ago). Climate forcing was obtained from Miocene simulations by GENESIS (Burls et al., 2021), with different $CO_2$ levels and ice sheet configurations, using a recently developed matrix interpolation routine (Berends et al., 2018). We find competing influences of the atmosphere, ocean and solid earth on transient Miocene AIS variability:

- Compared to a simpler glacial index method, as used e.g. in Stap et al. (2019), our matrix interpolation method incorporates two important ice-sheet-atmosphere feedbacks: the positive albedo-temperature feedback, and the negative ice-volume-precipitation feedback. The former tends to increase hysteresis in the $CO_2$-V relation and decrease the amplitude of transient AIS variability, while the latter has the opposite effect (Fig. 5). The albedo-temperature feedback is the strongest of the two, as in the 40-kyr transient reference simulation, the ice volume variability amplitude is 21% smaller than when a glacial index method is used.

- Differences between the geological reconstructions of Antarctic bedrock topography at the Oligocene-Miocene boundary (23-24 Myr ago) and the mid-Miocene (14 Myr ago) are mainly located in West-Antartica. The subsidence of land below sea level during the early- and mid-Miocene impedes WAIS build-up. Consequently, it diminishes hysteresis in the $CO_2$-ice volume above floatation (V) relation and reduces 40-kyr AIS variability by 10% in strength (Figs. 5,6). Hence, under equal atmospheric and oceanic forcing conditions, the amplitude of transient AIS variability becomes progressively smaller during the early- and mid-Miocene, due to glacial erosion.


- The importance of ice shelf formation is demonstrated by comparing simulations in which LGM basal melt rates are applied, to simulations in which ice shelf growth is prevented (Figs. 8,9). Ice shelves start to form around $CO_2$ levels of $\sim$530 ppm at the margins of Coates Land and Oates Land and at $\sim$360 ppm approximately simultaneously around all other margins. They continue to affect grounded ice volume until the forcing $CO_2$ concentration is increased to
$\sim$728 ppm in our 400-kyr transient simulation. The formation of ice shelves leads to a thickening of the hinterlying grounded ice, particularly of the WAIS, and consequently increases hysteresis in the $CO_2$-V relation. 40-kyr transient AIS variability is amplified by 19% in case of LGM basal melt rates - and 5% in the reference case - compared to the case without ice shelves.

In our simulations, the ice-sheet-atmosphere feedbacks and the evolution of bedrock topography, that is due mostly to glacial
erosion, hence tend to reduce the contribution of AIS changes to orbital timescale benthic $\delta^{18}$O variability during the Miocene. Conversely, the formation of ice shelves eases the gaining of ice volume, and increases this contribution. Realistic simulations of Miocene AIS evolution are left to future research.

*Code availability.* The code for IMAU-ICE v1.1.1-MIO is available from https://github.com/IMAU-paleo/IMAU-ICE/releases/tag/v1.1.1-MIO.

*Data availability.* All relevant data related to this article will be made openly accessible from the PANGAEA database (https://pangaea.de/).

## Appendix A: Ice-sheet model settings

In IMAU-ICE, Glenn's flow law exponent is set to $n = 3$, and the SIA- and SSA-based velocities are enhanced by factors 5 and 1 respectively. The basal stress is calculated using a Coulomb power law model for pseudo-plastic till (exponent $q = 0.3$ and threshold velocity $u_{threshold} = 100\,\text{m/s}$) (Bueler and Brown, 2009). The till friction angle, which determines the yield
stress, varies linearly between 5° for bedrock elevations of -1000 m and lower, and 20° for those of 0 m and above. The geothermal heat flux is taken from Shapiro and Ritzwoller (2004). A basic elastic lithosphere-relaxing asthenosphere (ELRA) model (Lingle and Clark, 1985; Le Meur and Huybrechts, 1996) is used to calculate bedrock adjustment, using a mantle density of $3300\,\text{kg}\,\text{m}^{-3}$, a lithospheric flexural rigidity of 1.0 x $10^{25}\text{kg}\,\text{m}^2\,\text{s}^{-2}$, and a relaxation time of 3000 yr. The sea level is kept constant at present-day level (0 m).

## Appendix B: Climate matrix forcing

Using a so-called matrix method (Pollard, 2010; Pollard et al., 2013), a time-continuous climate forcing for the ice-sheet model can be calculated. This is achieved by interpolating between the GENESIS time-slice climate simulations, based on the varying





$CO_2$ levels and ice coverage. The matrix interpolation routine used in this study is described in detail in Berends et al. (2018), and was previously deployed to simulate the evolution of the North-American, Eurasian, Greenland and Antarctic ice sheets during the past 3.6 Myr (Berends et al., 2019, 2021).

## B1 Surface air temperature

For the applied surface air temperature, first two weighing factors for $CO_2$ ($w_{CO2}$) and ice-sheet coverage ($w_{ice}$) are calculated.

$$w_{CO2} = \frac{CO_2 - CO_{2,cold}}{CO_{2,warm} - CO_{2,cold}}, \tag{B1}$$

where $CO_2$ is the applied $CO_2$ level, and $CO_{2,cold}$ and $CO_{2,warm}$ the low (280 ppm) and high (840 ppm) $CO_2$ levels. Weight factor $w_{ice}$ is based on scaling between local absorbed insolation of the forcing climates, accounting for the effect of surface whitening (darkening) by ice sheet advance (retreat) on temperature. In order to calculate $w_{ice}$, first the absorbed insolation $I_{abs}$ is calculated from the incoming insolation ($Q_{TOA}$) and internally calculated surface albedo ($\alpha$) for the modeled field ($I_{abs,mod}$), and the fields interpolated between ($I_{abs,cold}$, $I_{abs,warm}$):

$$I_{abs} = (1 - \alpha) * Q_{TOA}. \tag{B2}$$

Then, the weighing field for insolation ($w_{ins}$) is calculated:

$$w_{ins} = \frac{I_{abs,mod} - I_{abs,cold}}{I_{abs,warm} - I_{abs,cold}} \tag{B3}$$

Finally, the weighing field for ice-sheet coverage is calculated from the smoothed (using a Gaussian smoothing filter with a radius of 200 km) and average (over the ice-sheet model grid) $w_{ins}$, to account for local as well as regional effects:

$$w_{ice} = \frac{1}{7} w_{ins,smoothed} + \frac{6}{7} w_{ins,avg}. \tag{B4}$$

For each grid cell in the domain the final weighing factor $w_{tot}$ is calculated as:

$$w_{tot} = \epsilon_{CO2} * w_{CO2} + (1 - \epsilon) * w_{ice}. \tag{B5}$$

$\epsilon_{CO2}$ is the ratio between the effects of $CO_2$ and ice-sheet coverage, predetermined (in our reference simulation) on the basis of GENESIS simulations 3fumebi (840 ppm $CO_2$, full ice sheet) and 1nomebi (280 ppm $CO_2$, no ice sheet):

$$\epsilon_{CO2,ref} = \frac{\overline{T}_{3fumebi} - \overline{T}_{1fumebi}}{(\overline{T}_{3fumebi} - \overline{T}_{1fumebi}) + (\overline{T}_{1nomebi} - \overline{T}_{1fumebi})} \approx \frac{\overline{T}_{3fumebi} - \overline{T}_{1fumebi}}{\overline{T}_{3nomebi} - \overline{T}_{1fumebi}}, \tag{B6}$$

where $\overline{T}$ is the average temperature over the Antarctic domain. In the reference case $\epsilon_{CO2} = 0.62$. The weighing factors are clipped at -0.25 and 1.25, to avoid spurious extreme temperatures and precipitation.



Next, the reference temperature ($T_{ref}$) and surface height ($h_{ref}$) are obtained by interpolating between the cold ($T_{cold}, h_{cold}$) and warm ($T_{warm}, h_{warm}$) forcing:

$$T_{ref} = w_{tot} * T_{warm} + (1 - w_{tot}) * T_{cold} \tag{B7}$$

and likewise

$$h_{ref} = w_{tot} * h_{warm} + (1 - w_{tot}) * h_{cold}. \tag{B8}$$

The applied 2-m air temperature field $T_{applied}$ is obtained by first correcting for the height differences between the modeled ($h_{mod}$) and reference ($h_{ref}$) surface height using a uniform lapse rate $\lambda = 8\,\mathrm{K/km}$. Finally, a spatially-varying bias correction based on the difference between the modeled preindustrial temperature ($T_{PI}$) and the 1957-2002 ERA40 reanalysis data (Uppala et al., 2005) ($T_{ERA40}$) is applied:

$$T_{applied} = T_{ref} - \lambda * (h_{mod} - h_{ref}) + (T_{ERA40} - T_{PI}). \tag{B9}$$

## B2 Precipitation

Precipitation is weighed based on the total ice volumes modeled ($V_{mod}$) and in the cold ($V_{cold}$) and warm climates ($V_{warm}$):

$$w_{tot} = 1 - \frac{V_{mod} - V_{warm}}{V_{cold} - V_{warm}}. \tag{B10}$$

A reference precipitation field is obtained, by interpolating between the warm ($P_{warm}$) and cold ($P_{cold}$) states:

$$P_{ref} = \exp\left[w_{tot} * \log[P_{warm}] + (1 - w_{tot}) * \log[P_{cold}]\right]. \tag{B11}$$

Similarly as for the temperature, a bias correction is applied:

$$P_{ref,corr} = P_{ref} * \frac{P_{ERA40} + 0.008}{P_{PI} + 0.008}. \tag{B12}$$

The term 0.008 is added to avoid problems caused by division by very small numbers. And finally, the precipitation field is downscaled to the finer ice-sheet model grid, using a Clausius-Clapeyron relation (Lorius et al., 1985; Huybrechts, 2002; De Boer et al., 2013):

$$P_{applied} = P_{ref,corr} * \left(\frac{T_{I,ref}}{T_I}\right)^2 * \exp\left[22.47 * \left(\frac{273.16K}{T_{I,ref}} - \frac{273.16K}{T_I}\right)\right] \tag{B13}$$

with

$$T_I = 88.9 + 0.67 * T_{ref}, \tag{B14}$$

$$T_{I,ref} = 88.9 + 0.67 * (T_{ref} - \lambda * (h_{mod} - h_{ref})). \tag{B15}$$

*Author contributions.* LBS designed the research and performed the experiments, with technical assistance from CJB. LBS, CJB, MDWS and RSWvdW analysed the results. EGWG provided the GENESIS results used as climate forcing in this study. LBS drafted the paper, with input from all co-authors.



*Competing interests.* The authors declare that they have no conflict of interest.

*Acknowledgements.* L.B. Stap is funded by the Dutch Research Council (NWO), through VENI grant VI.Veni.202.031. C.J. Berends is supported by PROTECT, which has received funding from the European Union's Horizon 2020 research and innovation programme under
grant agreement no. 869304. We thank our colleague Peter Bijl for commenting on an earlier draft of this article. We further thank the researchers that reconstructed the Antarctic paleotopographies and paleobathymetries for making them publicly available, and in particular Isabel Sauermilch for providing them to us.



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



**Table 1.** Name and description of the experiments carried out. Each experiment comprises 15 simulations, as explained in the main text and listed in Table 2.

| Simulation name | Description |
|---|---|
| REF | Default settings (see Sect. 2) |
| SMB | Increased ablation: $c_3 = -0.25\,\mathrm{m\,yr^{-1}}$ |
| INSOL | Interpolation between forcing climates with low (1fucobi) and high summer insolation (3nowabi) |
| FEEDB | Interpolation of climate forcing only based on $CO_2$ |
| FEEDB onlyT | Interpolation of temperature only based on $CO_2$ |
| FEEDB onlyP | Interpolation of precipitation only based on $CO_2$ |
| TOPO 14Ma | Bedrock topography pertaining to 14 Myr ago (Hochmuth et al., 2020a; Paxman et al., 2019) |
| TOPO 34Ma | Bedrock topography pertaining to 34 Myr ago (Hochmuth et al., 2020a; Paxman et al., 2019) |
| TOPO PD | Present-day bedrock topography (Morlighem et al., 2020), isostatically rebounded after ice sheet removal |
| TOPO Wilson_max | Bedrock topography pertaining to Late Eocene (Wilson et al., 2012), max reconstruction |
| TOPO Wilson_mean | Bedrock topography pertaining to Late Eocene (Wilson et al., 2012), mean of min and max reconstructions |
| BMB LGM | LGM ocean conditions ($M_{expo} = 0\,\mathrm{m/yr}$, $M_{deep} = 2\,\mathrm{m/yr}$, and $T_w = -5°\mathrm{C}$) |
| BMB no_shelves | Constant sub-shelf melt rate $M = 400\,\mathrm{m/yr}$ |

**Table 2.** Description of the 15 simulations spanning each experiment.

| Type | Initial conditions | $CO_2$ levels | Runtime (model years) |
|---|---|---|---|
| Steady-state | Miocene topography, no ice | 280, 392, 504, 616, 728, and 840 ppm | 150 kyr |
| Steady-state | 280-ppm steady-state IMAU-ICE run | 392, 504, 616, 728, and 840 ppm | 150 kyr |
| Transient | 840-ppm steady-state IMAU-ICE run | linearly from 840 to 280 to 840 ppm | 100, 400, and 200 (5x40) kyr |
| Transient | 280-ppm steady-state IMAU-ICE run | linearly from 280 to 840 to 280 ppm | 200 (5x40) kyr |



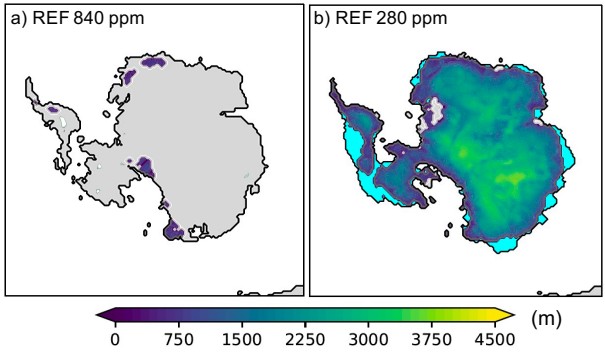

**Figure 1. (a)** Simulated equilibrated ice thickness from experiment REF with 840-ppm, and **(b)** 280-ppm $CO_2$ forcing, starting without ice.

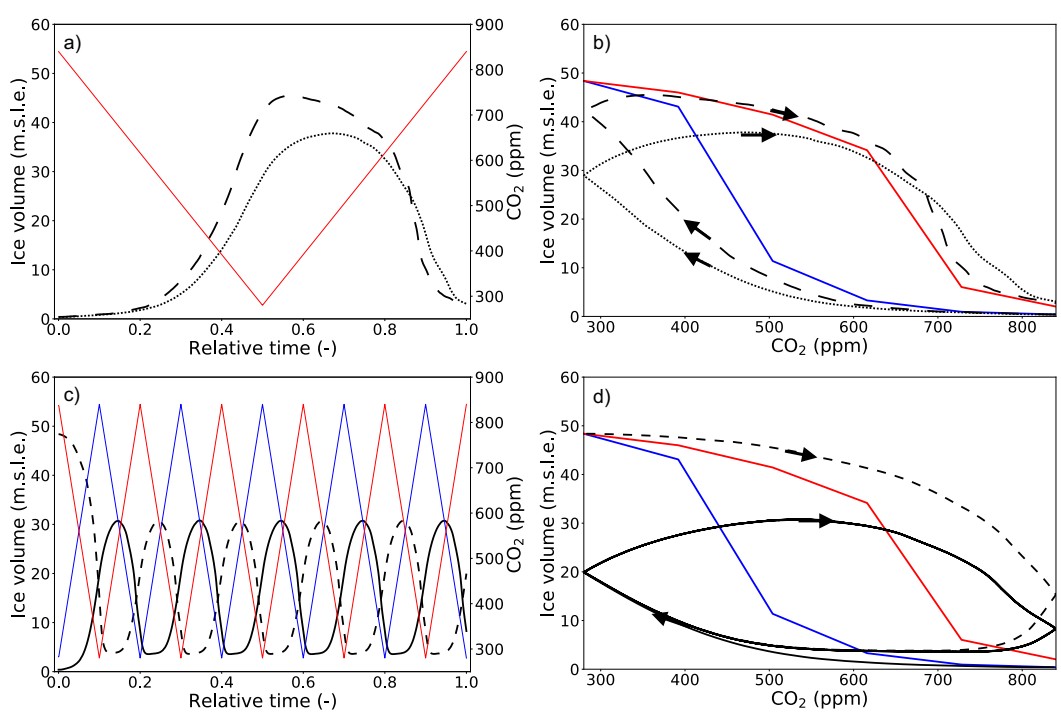

**Figure 2. (a)** Evolution of forcing $CO_2$ levels (red) and ice volume above floatation (in meters sea level equivalent; black) over time (relative to the length of the simulation), for the transient 100-kyr (dotted) and 400-kyr (long-dashed) REF simulations. **(b)** Relation between $CO_2$ and equilibrium ice volume (ascending branch, blue; descending branch, red), and transient ice volume. **(c)** and **(d)** Same for the regular ($CO_2$ red, ice volume solid black) and reversed ($CO_2$ blue, ice volume dashed black) transient 40-kyr simulations. Arrows in **(b)** and **(d)** indicate the progression direction of ice volume. This direction is similar (lower branch ascending, upper branch descending) in further figures.



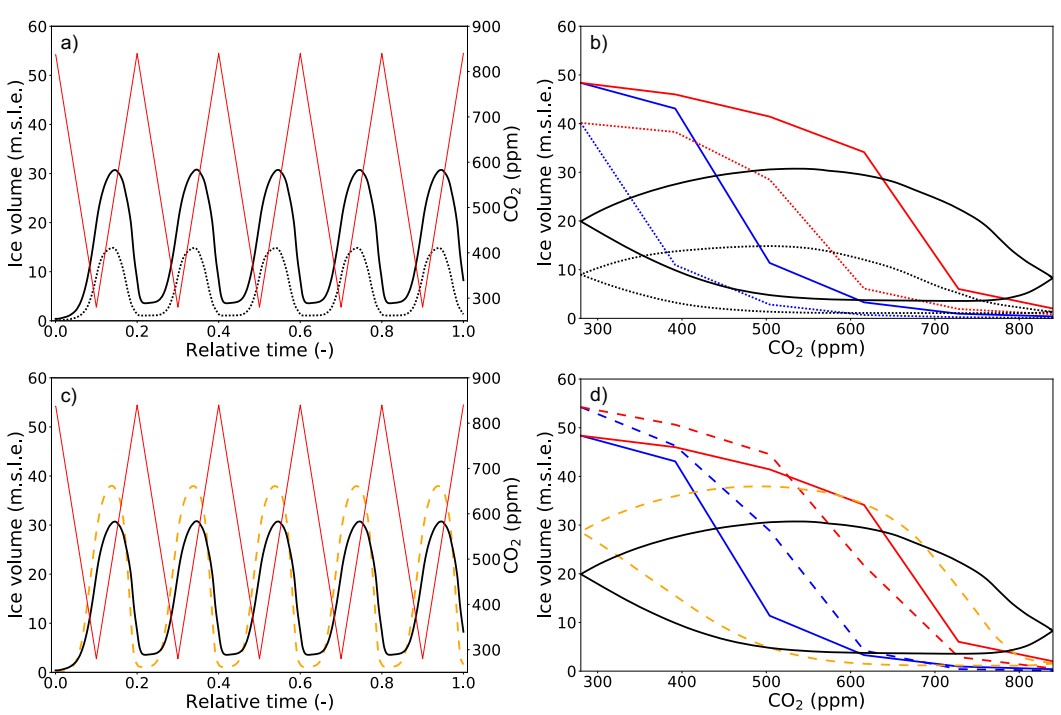

**Figure 3. (a)** Evolution of forcing $CO_2$ levels (red) and ice volume (black) over relative time, for the transient 40-kyr REF (solid) and SMB (dotted) simulations. **(b)** Relation between $CO_2$ and equilibrium ice volume (REF, solid; SMB, dotted), and transient ice volume. Only the equilibrium cycle (final 40 kyr) is shown. **(c)** and **(d)** Same for the REF (solid, black) and INSOL (dashed, orange) transient 40-kyr simulations.





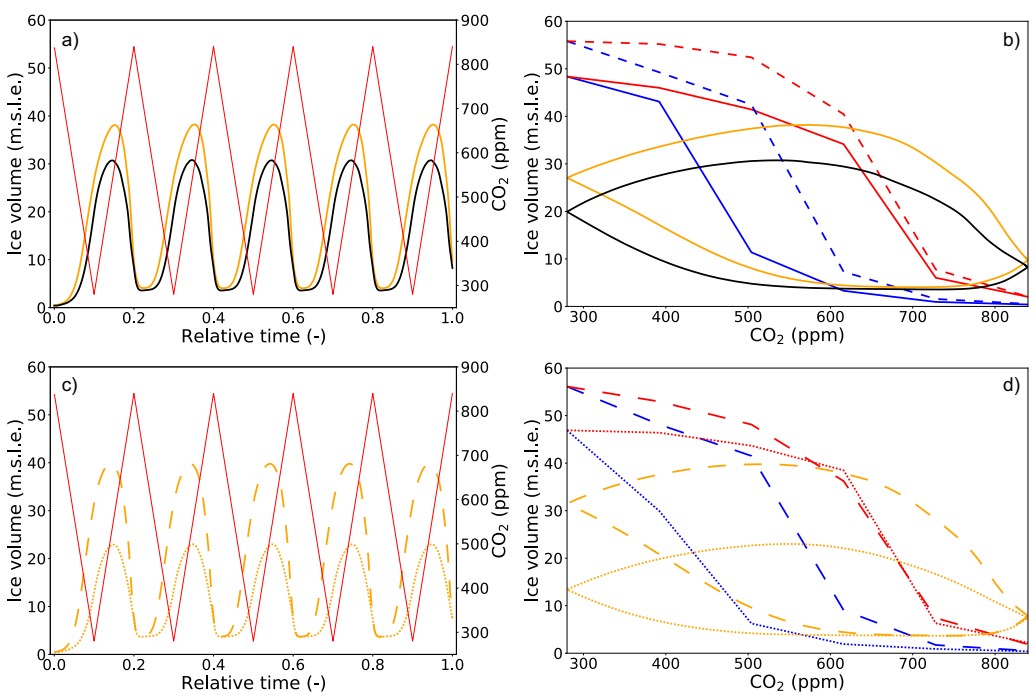

**Figure 4. (a)** Evolution of forcing $CO_2$ levels (red) and ice volume over relative time, for the transient 40-kyr REF (black) and FEEDB (orange) simulations. **(b)** Relation between $CO_2$ and equilibrium ice volume (REF, solid; FEEDB, dashed), and transient ice volume. Only the equilibrium cycle (final 40 kyr) is shown. **(c)** and **(d)** Same for the FEEDB onlyP (dotted) and FEEDB onlyT (long-dashed) transient 40-kyr simulations.





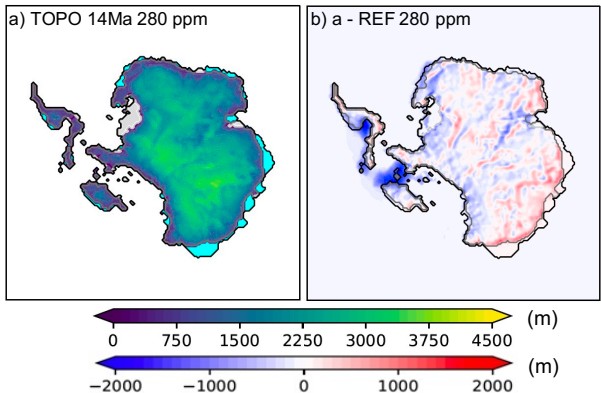

**Figure 5. (a)** Simulated equilibrated ice thickness from experiment TOPO 14Ma with 280-ppm $CO_2$ forcing. **(b)** Difference in ice thickness between TOPO 14Ma and REF with 280-ppm $CO_2$ forcing (grounding line and continental edge as in **(a)**).





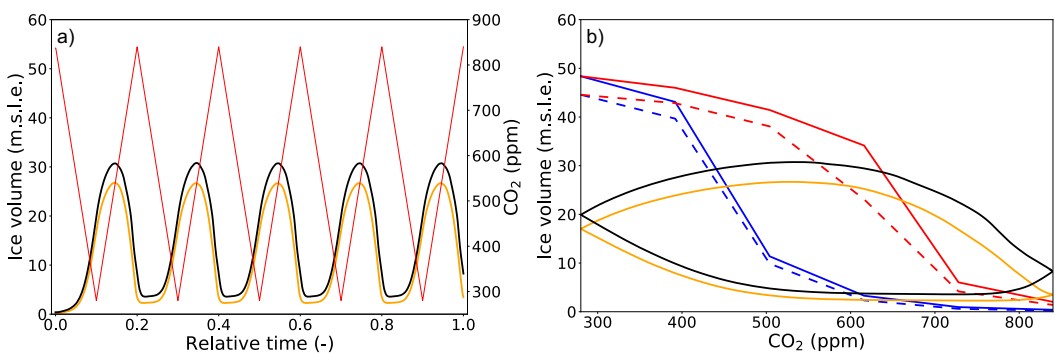

**Figure 6. (a)** Evolution of forcing $CO_2$ levels (red) and ice volume over relative time, for the transient 40-kyr REF (black) and TOPO 14Ma (orange) simulations. **(b)** Relation between $CO_2$ and equilibrium ice volume (REF, solid; TOPO 14Ma, dashed), and transient ice volume. Only the equilibrium cycle (final 40 kyr) is shown.





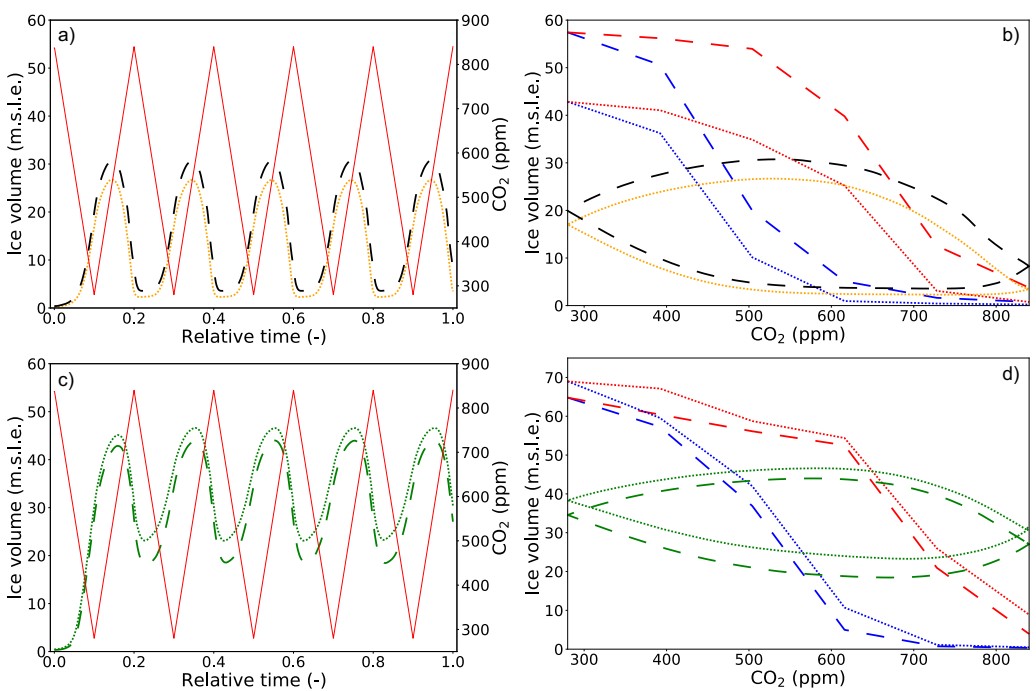

**Figure 7. (a)** Evolution of forcing $CO_2$ levels (red) and ice volume over relative time, for the transient 40-kyr TOPO 34Ma (black, long-dashed) and TOPO PD (orange, dotted) simulations. **(b)** Relation between $CO_2$ and equilibrium ice volume (TOPO 34Ma, long-dashed; TOPO PD, dotted), and transient ice volume. Only the equilibrium cycle (final 40 kyr) is shown. **(c)** and **(d)** Same for the TOPO Wilson_mean (green, dotted) and TOPO Wilson_max (green, long-dashed) transient 40-kyr simulations. Mind the differing y-axis scales in panel **(d)**.



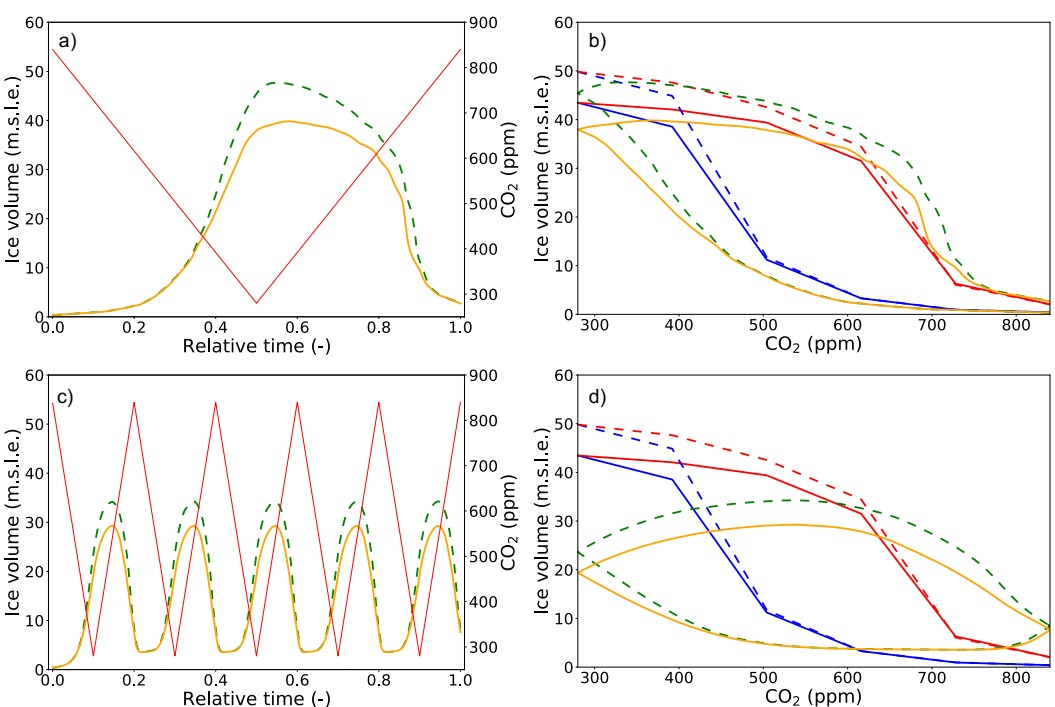

**Figure 8. (a)** Evolution of forcing $CO_2$ levels (red) and ice volume over relative time, for the transient 400-kyr BMB no_shelves (solid orange) and BMB LGM (dashed, green) simulations. **(b)** Relation between $CO_2$ and equilibrium ice volume (BMB no_shelves, solid; BMB LGM, dashed), and transient ice volume. **(c)** and **(d)** Same for the transient 40-kyr simulations. Only the equilibrium cycle (final 40 kyr) is shown.



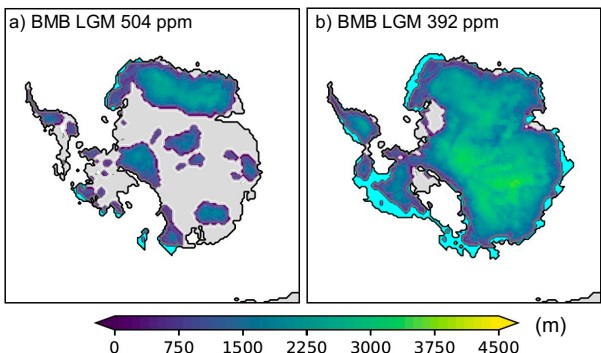

**Figure 9. (a)** Simulated equilibrated ice thickness from experiment BMB LGM with 504-ppm, and **(b)** 392-ppm $CO_2$ forcing, starting without ice.