# Peer review of "Net effect of ice-sheet-atmosphere interactions reduces simulated transient Miocene Antarctic ice sheet variability"

_The Cryosphere, 2021_

## Author Response (AR1)

**Reply to referees**

*Color coding:*
Black – comments by the reviewer (line numbers refer to initial submission)
Blue – reply by the authors (line numbers refer to revised manuscript with changes indicated)

**Reply to Anonymous Referee #1**

Stap et al. present numerical experiments of the Antarctic evolution during the mid- Miocene. They use a climate matrix interpolation scheme from GCM outputs to force their ice sheet simulations. They present equilibrium and idealised transient simulations and discuss a few feedbacks.

The methodology is sound and the numerical experiments well chosen. However there are several points that could/should be improved:

We thank the reviewer for a thorough examination of our work. We are pleased they agree with our set-up and appreciate our experiments. Below, we address their comments and indicate which actions we will take to revise the manuscript.

- Paper objective. The title is about the competing influence of the ocean, the atmosphere and the solid Earth and indeed the three aspects are discussed in the paper. However, it is not clear at the end of the introduction which are the main questions related to these processes and the added value of the present study. The literature (at least for the numerical part) is correctly acknowledged (not necessarily in the introduction) but it is not clear yet how the paper builds up from this. For example, the authors present three major improvements with respect to earlier works: 1- updated bedrock topography; 2- transient experiment and; 3- ice sheet – climate interactions accounted for. However, these are not always discarded in the recent literature (e.g. Halberstadt et al., 2021; Colleoni et al., 2018; Paxman et al., 2020). There is a lot of material in the paper but what can we learn from it can be more precisely described.

The title of the manuscript will be adjusted to '**Net effect of ice-sheet-atmosphere interactions reduces simulated transient Miocene Antarctic ice sheet variability'** to reflect a stronger focus on the effects of ice-sheet-atmosphere interactions. In the last paragraph of the introduction, we will indicate more clearly how our work improves upon earlier research. Compared to Gasson et al. (2016), Colleoni et al. (2018), Paxman et al. (2020) and Halberstadt et al. (2021), we study transient variability in addition to steady-state results. Compared to De Boer et al. (2010) and Stap et al. (2019, 2020), we include ice-sheet-atmosphere inter-actions and use an updated bedrock topography. Combined with a more verbose outline of the experiments performed and their purpose, this serves to clarify the intent of our study.

*Changes: title, and pages 3-4 lines 66-93.*

- Description of the methods. I would suggest to add a section on the matrix interpolation method after sec. 2.2 but before 2.3. Something simple that explains the main principle, keeping all the details in the appendix. I would like to read the rationale behind eq. B6 (insolation weighing). Also, climate model description section should be rewritten: it is not clear what existing simulations did you use ("nowabi" is not mentionned here) and the style can be largely improved (3 line parenthesis with embedded parenthesis is not easy to read).

As suggested by the reviewer, we will include a brief description of the matrix interpolation method in the methods section right after Sect. 2.2. In the revised manuscript, we discard the presentation of our experiment INSOL, because this does not contribute to the main story line and only confused the reviewers. Therefore, now in all our experiments GENESIS simulation 1fumebi (280 ppm, large ice sheet) provides the 'cold' forcing, and 3nomebi (840 ppm, no ice sheet) provides the 'warm' forcing. These GENESIS simulations are introduced in a clearer manner, with fewer parentheses, in the revised method section.

*Changes: pages 6-7 lines 164-186.*

We will also explain the rationale behind Equation B6, which we use to determine the ratio between the effect of externally prescribed $CO_2$ changes and internally calculated ice-sheet coverage on the interpolated forcing temperature. To this end, we use GENESIS simulations 3fumebi (840 ppm $CO_2$, large ice sheet) and 1nomebi (280 ppm $CO_2$, no ice sheet), which isolate the separate effects of $CO_2$ and ice-sheet coverage respectively. The (spatially averaged) relative influence of $CO_2$ on temperature is then calculated as the isolated effect of $CO_2$ divided by the combined influence of $CO_2$ and ice-sheet coverage (Eq. B6).

*Changes: page 19 lines 587-592.*

- Results, shelves (particularly p. 10-11 and p.12 l. 386-390). You use a highly parametrised sub-shelf melt model that does not depend on the climatic simulations. If I understand correctly it is the result of an ad-hoc combination of linear functions of CO2. I understand the need for something simple and do not think that this is a critical flaw in the paper, however it prevents you for providing insights on the different thresholds for ice shelves formation and stability.

We agree with the reviewer that the thresholds for ice shelf formation and stability that we find, are not to be considered robust quantifications. Therefore, we will refrain from presenting them as such in the revised manuscript. Instead, throughout the revised manuscript, we present the experiments using different basal melt rates (BMB LGM, BMB no_shelves) as essential sensitivity tests. They serve to quantify where and under which conditions ice shelves form and perish in our simulations, and how they influence the variability of the grounded ice volume. According to this different assessment of these experiments, we will adjust the title of the manuscript, as well the abstract and conclusion section where these results will not be mentioned anymore.

*Changes: title, page 1 lines 17-20 (deletion), and page 17 lines 516-523 (deletion).*

- Results presentation. You present multiple times the CO2/volume and hysteresis curves (6 out of 9 figures). I think you should explain better what can you learn from such a figure when you first show it. The style of the figure can also be improved and you will find a few suggestion in the "specific comments" section. It would also be very nice to show what the simulated ice sheets actually look like at different moment of the hysteresis (for a given volume I assume that the ice sheets of the ascending and descending branches do not look the same).

The hysteresis figures can seem a bit convoluted at first sight, with multiple steady-state and transient results. However, we think it is necessary that this information is contained in a single figure, to allow for a proper comparison between the different results. We will aim to improve the readability of these figures in the following ways:

- The equilibrium results are highlighted by diamonds.
- The evolution of $CO_2$ is indicated by pink and purple instead of red and blue lines in the a- and c-panels, so that it is clearly distinguishable from the equilibrium results in the b- and d-panels.
- Arrows are added to the transient results in all b- and d-panels, to indicate the progression direction.
- Legends are included to indicate which experiments are displayed.
- All the different symbols and lines are described in the figure captions.

*Changes: Figures 1, 3, 5, 6, S5 and S8.*

Furthermore, we will guide the reader on how to read these hysteresis figures, upon their first occurrence when the REF experiments are described (new Fig. 1). Finally, we will include a supplementary figure with 'intermediate' states, i.e. the results of the steady-state 392-504-, 616-, and 728-ppm $CO_2$ REF simulations of both the ascending and descending branch.

*Changes: Figure S4, and pages 8-9 lines 235-244.*

- Methods. You use a bias correction based on present-day bias with respect to ERA40. I have two questions related to this: 1- why ERA40, which is an old dataset now, and not a regional climate model (such as RACMO or MAR) as the reanalysis generally does not perform well for southern high latitudes? 2- Does it really make sense to use the same bias correction also for the climate simulations in which there is no ice in Antarctica? The effect of topography could eventually change drastically the wave pattern and associated biases.

Similar to Stap et al. (2019), we use ERA40 reanalysis data for a bias correction. This is because that was the climate used to benchmark the different ice sheet models contributing to PLISMIP-ANT (De Boer et al., 2015). For the reasons mentioned by the reviewer, it may be more appropriate to use regional climate model results, or refrain from using a bias correction altogether. We will consider this for future research. However, for the current study, we submit that the ice volume differences are determined for the large part by the difference between the warm and cold forcing climates as well as the value of the ablation parameter (c3 in Eq. 6).

In the last paragraph of the revised discussion section, we will discuss the purpose of the bias correction, which is to obtain a more detailed reference climate, as well as the fact that it introduces a small but significant warm bias because we use a present-day rather than pre-industrial reference climate (also noted in Stap et al., 2019).

*Changes: pages 15-16 lines 478-482.*

- I am not entirely convinced that you should use the terminology "volume-precipitation feedback". There is a negative feedback associated with precipitation but I would say that it is more related to large-scale temperature. As a first approximation, colder air contains less humidity even though the process is more complex. Your have a volume-precipitation feedback because it is parametrised this way. I guess you could have written Eq. B10 using the ice extent instead of volume (you assume that precipitation change is more related to ice volume change, you did not demonstrate it). Alternatively, you could have use a local correction as for insolation to account for precipitation changes (using surface height for example).

Indeed, we called it 'ice-volume-precipitation feedback' because it is parameterised that way in our model. In the revised manuscript we will explain that this represents our manner of capturing the ice-sheet desertification effect, i.e. the reduction of snow deposition on a growing ice sheet. We will refer to it as *a,* rather than *the*, negative feedback between ice volume and precipitation.

*Changes: page 1 lines 9-10, and page 7 lines 184-186.*

- Try to be more specific instead of speaking of "warm/cold". You have three factors that can influence this warm/cold definition: CO2 concentration, orbital parameters and topography/ice mask in the climate model. This can be confusing sometimes in the text. Check and be consistent throughout the manuscript.

We interpolate the forcing temperature and precipitation between two climates, a 'cold' and a 'warm' one. Because we no longer present the results of simulation INSOL in the revised manuscript (see above), 'cold' now always refers to a large ice sheet and low $CO_2$ level, provided by GENESIS simulation 1fumebi. 'Warm' refers to no ice, and a high $CO_2$ level, provided by GENESIS simulation 3nomebi. Both simulations use present-day orbital settings. This will be clarified in the method section of the revised manuscript.

*Changes: pages 6-7 lines 171-178.*

- Introduction. A justification for the cyclicity at 40k, 100k and 400k would be nice.

We will indicate that these are quasi-orbital timescales.

*Changes: page 3 line 80.*

- Equations. Please write the x,y dependency when appropriate.

The x,y-dependencies will be included in all equations.

*Changes: all equations.*

- Since albedo is an important driver for your ice sheet evolution, please provide more details on its implementation in IMAU-ICE.

In the method section of the revised manuscript, we include a description of the albedo scheme used in IMAU-ICE. A background albedo is first calculated: 0.1 for ocean-covered grid cells, 0.2 for soil, and 0.5 for ice. On top of that, the effect of snow, with an albedo of 0.85, is added. This effect depends on the transiently changing snow layer and ablation of the previous year.

*Changes: page 6 lines 150-156.*

*Specific comments*

The manuscript will be revised along the lines of these comments, as indicated below.

- l. 9. Related to one of my previous comment, I would suggest rephrase this in the abstract since the ice volume-precipitation feedback sounds unconventional.

In the revised abstract, we rephrase this as 'a negative feedback between ice volume and precipitation'. Please see our reply to the general comment on this issue above.

- l. 22. But with drastically different timescales.

This will be mentioned.

*Changes: page 2 lines 23-24.*

- l. 81. Specific treatment of the grounding line?

In the version of IMAU-ICE we use for this study, no additional grounding-line parameterisations are used.

*Changes: page 4 lines 102-103.*

- l. 103. Why?

The only calving routine implemented in IMAU-ICE is thickness calving, i.e. calving floating ice with a thickness below a certain threshold. We exclude this in our simulations, because it could hinder the growth of new ice shelves (e.g. Ritz et al., 2001).

*Changes: page 5 lines 127-129.*

- l. 109. These are probably computed on the fine resolution grid. How do you do the downscaling of the climate fields from the T31 to the 40km grid?

The GENESIS fields are remapped using bilinear interpolation based on longitude and latitude. Longitude and latitude coordinates are assigned to the ice-sheet model grid using an inverse oblique stereographic projection (Reerink et al., 2010).

*Changes: page 4 lines 105-106, and page 5 lines 140-141.*

- l. 137-138. Why do you keep the "bi" in the manuscript if it is not relevant?

The GENESIS simulations were introduced in Burls et al. (2021). To maintain congruity between articles, we use exactly the same nomenclature here.

- l. 137-138. Even in the "nome" simulations? Vegetation is not allowed to grow when ice is absent? Clarify this in the text. I would have assumed that vegetation is an important feedback for ice sheet inception.

The use of BIOME4 in the forcing climate simulations affects the vegetation on non-glaciated land and hence the climate.

*Changes: pages 6-7 lines 176-178.*

- l. 152. If this figure is cited first, it should be numbered Fig. 1.

It will be Fig. 1 in the revised manuscript.

- l. 156-160. I found the description of the transient simulations unclear.

The description of the transient simulations will be rephrased in a clearer manner.

*Changes: pages 7-8 lines 205-212.*

- l. 168. The present-day Antarctic ice sheet is the result of a deglaciation so it is not that bad if the volume obtained after an equilibrium is not in agreement with the present-day volume.

Indeed, these PI-reference simulations are just meant as a minimal check for model performance.

- l. 172-173. I might not understand you here: Reese et al. (2018) discuss the impact of the buttressing on the grounded part of the ice sheet. Also the ABUMIP project has shown a strong impact of buttressing for the present-day ice sheet (most ice sheet models in ABUMIP looses almost all the WAIS in the abuk experiments).

We accidentally referred to the wrong paper, it should be:

*Reese, R., Gudmundsson, G. H., Levermann, A., and Winkelmann, R.: The far reach of ice-shelf thinning in Antarctica, Nature Climate Change, 8, 53–57, https://doi.org/10.1038/s41558-017-0020-x, 2018b.*

This article discusses which parts of the ice shelves are the most important for buttressing the grounded ice (see e.g. their Fig. 1). From their results, which generally show a smaller influence of the parts of ice shelves that are furthest away from the grounded ice, we gather that overestimating the ice shelf area has a relatively small effect on grounded ice flow.

*Changes: page 8 line 226.*

- l. 187- 191. It would be nice to have maps here. As it stands it is very descriptive.
- l. 191-192. Unclear.

Rather than including extra figures, we will point out that the reader can assess the growth and decay rates from the slope of the ice volume progression in both the a- and b-panels of Fig. 2 (which will be Fig. 1 in the revised manuscript).

*Changes: page 9 line 150.*

- l. 200-201. "…we now interpolate…" how? What equation is used? You replace the "nomebi" model outputs by the "nowabi" model outputs?
- l. 201. You presented the "nomebi" as warm Antarctic summer (l. 135). "nowabi" was never presented before.
- l. 204-205. Does "nowabi" warmer than "nomebi"? Why do you obtain larger ice volume then? It seems that it is simply an artefact due to a different epsilonCO2 that gives more importance to the cold state.
- l. 207-208. What is your point here? And why this is observed?
- l. 208-209. Remove this sentence?

From the comments above, as well as from the comments of reviewer #2, it has become clear to us that the discussion of the results of experiment INSOL caused confusion. Since they were only meant a side note, and do not contribute to the main story line, we have chosen to refrain from discussing them in the revised manuscript. Therefore, this whole section will be removed.

*Changes: page 9 lines 260-276 (deletions).*

- l. 230-231. If I understand correctly, you are saying that ablation change is more efficient than precip change to explain ice volume change. I guess that this is not particularly surprising and have been shown earlier in different context. But I might be overlooking something?

We quantify the effects of the *feedbacks* between temperature and ice volume, and between precipitation and ice volume. We do that by comparing runs using the matrix method, to runs where the temperature and precipitation forcing is interpolated based on the prescribed $CO_2$ level alone. In both cases, using the matrix and index method, precipitation and temperature can change during the simulations. We do not head-on investigate the separate effects of

precipitation and temperature on ice volume change. That would require running the model with only temperature or precipitation changes from the climate simulations, as was done in Stap et al. (2019).

- l. 230-231. You have chosen a c3 parameter that maximises the hysteresis. You might have ended up with slightly different conclusions with a smaller c3 parameter?

Please note that we did not tune the c3 parameter to maximise hysteresis. Rather, our REF settings yield an ice sheet similar in extent to the results of Stap et al. (2019) and DeConto and Pollard (2003) at 280 ppm, which disappears almost completely at 840 ppm. This ensures that we can most effectively study large-scale variability of the East-Antarctic ice sheet. A different tuning would shift the $CO_2$ at which these ice sheet sizes are simulated, but does not (qualitatively) affect the shape of the $CO_2$-ice volume relation.

*Changes: page 9 lines 265-268.*

- l. 237-238. Consider adding a map of bedrock topography difference in Fig. 5. Or alternatively a map of the various bedrock topographies would be nice for Sec. 3.3.

We will include a figure showing all the different bedrock topographies used.

*Changes: Figure S6.*

- l. 293. Halberstadt et al. (2021) use a more sophisticated surface mass balance scheme. The might have a better representation of orographic precipitation for example?

We use an insolation temperature melt (ITM) scheme, accounting for the direct effect of insolation on surface melt, whereas they deploy a positive degree day (PDD) scheme. Additionally, they use a regional climate model for downscaling the forcing, which indeed allows for capturing orographic precipitation in more detail. We instead use a bias correction to obtain a more detailed reference climate.

*Changes: page 15 lines 473-479.*

- l. 318. I am not sure you can really quantify this since you use a very simple ocean-ice interaction representation in your modelling setup.

We would like to note that the $CO_2$ level at which the ice sheet first reaches the ocean does not depend on ocean temperatures in our set-up. Since up until that point, the ice is not in contact with the ocean, this $CO_2$ level is solely dependent on the atmosphere forcing and ice dynamics. In the revised manuscript, we will nonetheless make it clear that we are discussing the impact of ocean forcing on ice volume in *our* results, and that we do not draw general conclusions from this. The title of the revised manuscript, as well as the abstract and conclusion will also reflect that.

*Changes: title, and pages 3-4 lines 66-93.*

- l. 408-409. Why? Did you asses the sensitivity of your results to sea level change / value?

The sea level is kept constant at present-day level (0 m), because local Antarctic sea level changes are very uncertain during the Miocene as they are mainly due to Antarctic ice sheet changes. Local sea level changes can differ substantially from the global mean (e.g. Stocchi et al., 2013), and their calculation would require solving the sea-level equation which is not yet available in IMAU-ICE. We have performed three tests, applying a constant 7 m higher sea level (mimicking the absence of the Greenland ice sheet), as well as applying internally calculated global mean, and minus the global mean, sea level changes. These tests have shown negligible differences in comparison to our reference results. They will be briefly discussed, but now shown, in the revised manuscript.

*Changes: page 19 lines 545-551.*

- l.424-428. Iabs is computed from the albedo given by the ice sheet model internal routine. But then how Eq. B3 works? Is the albedo kept identical in "mod" and for "cold"/"warm", only changing the value of QTOA? Please provide a bit more details here.

In order to obtain $w_{ice}(x,y)$, first the absorbed insolation $I_{abs}(x,y)$ is calculated from the incoming insolation and internally calculated surface albedo. This is done every 10-yr coupling time step for the interpolated modeled field. For the cold (1fumebi) and warm (3nomebi) GCM snapshot fields, this is done at the start of each simulation by running the SMB model (which is more extensively described in the revised manuscript), without ice dynamics, for 10 yrs to determine the albedo. Incoming insolation is kept constant at the present-day values, so that differences between the cold, warm and modeled $I_{abs}(x,y)$ fields are solely determined by the albedo.

*Changes: Pages 18-19 lines 568-574.*

- l. 428. cold/warm refers to the CO2 level? This is confusing since you have three factors for warm/cold: CO2 concentration, the orbital parameters and the topography/ice mask in the climate model.

As indicated above, 'cold' now always refers to a large ice sheet and low $CO_2$ level, provided by GENESIS simulation 1fumebi, and 'warm' refers to no ice and a high $CO_2$ level, provided by GENESIS simulation 3nomebi.

- l. 431. Eq. B4: do you have any justification for this 1:7 ratio? Any number would work the same?

The viability of the (1:7) partitioning $w_{ins,smoothed}(x,y)$ and $w_{ins,avg}$ was demonstrated by Berends et al. (2018), who simulated ice-sheet evolution over the last glacial cycle, showing that model results agree well with available data in terms of ice-sheet extent, sea-level contribution, ice-sheet surface temperature and contribution to benthic $\delta^{18}O$. We deem a sensitivity analysis with respect to this setting to be beyond the scope of the current study.

*Changes: page 19 lines 582-584.*

- l. 436. Please provide a literal justification of Eq. B6. You could drop the "ref" on the left hand side of the equation since you computed this only once right?

*Please see our response to the general comment of the reviewer on this issue. Since we no longer discuss simulation INSOL in the revised manuscript, 'ref' can and will be dropped.*

*Changes: page 19 lines 588-595.*

- l. 437. Insert a new line here from "The weighing..." since this applies to wice and wco2 (not epsilonCO2) if I am not wrong?

*Weighing factors $w_{CO2}$ and $w_{ins}(x,y)$ are clipped at -0.25 and 1.25, to avoid spurious extreme temperatures and precipitation. This will be mentioned right after Eq. B3.*

*Changes: page 19 line 577.*

- Fig. 1. Highlight the points corresponding to the equilibrium simulations (stars, big dots, anything).
- Fig. 1. Try to avoid using the same colour for different things (red in a and b). Add a better description of the different coloured lines in each sub-panel. For example: we have to guess that the black dotted and dashed in b) is the same as in a) when the red continuous represents something else in a). In d) it seems that you have two ascending branches for the simulation "regular CO2" (solid black line). "Regular" transient simulation sounds weird.
- Fig. 3. In d) what are the dashed red and blue lines?
- Fig. 4. In b) the orange is transient for FEEDB while the dashed red/blue is equilibrium counterpart? This is really confusing in my opinion.

*Fig. 1 and similar 'hysteresis' figures (i.a. Figs. 3, 4) will be altered, as described above in our reply to the general comment of the reviewer on this issue.*

*Changes: Figures 1, 3, 5, 6, S5 and S8.*

- Fig. 5 and Fig. S2 / S4. The colour gradient for the difference plot is hard to read, it mostly gives an indication on the signs. You could try with a discontinuous colour scheme or with isocontours?

*We will use a colour scheme that more clearly shows the differences.*

*Changes: Figures 4, S3, and S9.*

**REFERENCES:**

Berends, C. J., de Boer, B., and Van de Wal, R. S. W.: Application of HadCM3@Bristolv1.0 simulations of paleoclimate as forcing for an ice-sheet model, ANICE2.1: set-up and benchmark experiments, Geoscientific Model Development, 11, 4657–4675, https://doi.org/10.5194/gmd-11-4657-2018, 2018.

Burls, N. J., Bradshaw, C. D., De Boer, A. M., Herold, N., Huber, M., Pound, M., Donnadieu, Y., Farnsworth, A., Frigola, A., Gasson, E., et al.: Simulating Miocene warmth: insights from an opportunistic multi-model ensemble (MioMIP1), Paleoceanography and Paleoclimatology, p. e2020PA004054, https://doi.org/10.1029/2020PA004054, 2021.

Colleoni, F., De Santis, L., Montoli, E., Olivo, E., Sorlien, C. C., Bart, P. J., Gasson, E. G. W., Bergamasco, A., Sauli, C., Wardell, N., et al.: Past continental shelf evolution increased Antarctic ice sheet sensitivity to climatic conditions, Scientific Reports, 8, 1–12, https://doi.org/10.1038/s41598-018-29718-7, 2018.

De Boer, B., Van de Wal, R. S. W., Bintanja, R., Lourens, L. J., and Tuenter, E.: Cenozoic global ice-volume and temperature simulations with 1-D ice-sheet models forced by benthic $\delta^{18}O$ records, Annals of Glaciology, 51, 23–33, 2010.

De Boer, B., Dolan, A. M., Bernales, J., Gasson, E., Goelzer, H., Golledge, N. R., Sutter, J., Huybrechts, P., Lohmann, G., Rogozhina, I., Abe-Ouchi, A., Saito, F., and van de Wal, R. S. W.: Simulating the Antarctic ice sheet in the late-Pliocene warm period: PLISMIP-ANT, an ice-sheet model intercomparison project, The Cryosphere, 9, 881–903, https://doi.org/10.5194/tc-9-881-2015, 2015.

DeConto, R. M. and Pollard, D.: Rapid Cenozoic glaciation of Antarctica induced by declining atmospheric $CO_2$, Nature, 421, 245–249, https://doi.org/10.1038/nature01290, 2003.

Gasson, E., DeConto, R. M., Pollard, D., and Levy, R. H.: Dynamic Antarctic ice sheet during the early to mid-Miocene, Proceedings of the National Academy of Sciences, 113, 3459–3464, https://doi.org/10.1073/pnas.1516130113, 2016.

Halberstadt, A. R. W., Chorley, H., Levy, R. H., Naish, T., DeConto, R. M., Gasson, E., and Kowalewski, D. E.: CO2 and tectonic controls on Antarctic climate and ice-sheet evolution in the mid-Miocene, Earth and Planetary Science Letters, 564, 116908, https://doi.org/10.1016/j.epsl.2021.116908, 2021.

Paxman, G. J. G., Gasson, E. G.W., Jamieson, S. S. R., Bentley, M. J., and Ferraccioli, F.: Long-Term Increase in Antarctic Ice Sheet Vulnerability Driven by Bed Topography Evolution, Geophysical Research Letters, 47, e2020GL090003, https://doi.org/10.1029/2020GL090003, 2020.

Reerink, T. J., Kliphuis, M. A., and van deWal, R. S.W.: Mapping technique of climate fields between GCM's and ice models, Geoscientific Model Development, 3, 13–41, https://doi.org/10.5194/gmd-3-13-2010, 2010.

Ritz, C., Rommelaere, V., and Dumas, C.: Modeling the evolution of Antarctic ice sheet over the last 420,000 years: Implications for altitude changes in the Vostok region, Journal of Geophysical Research: Atmospheres, 106, 31 943–31 964, https://doi.org/10.1029/2001JD900232, 2001.

Stap, L. B., Sutter, J., Knorr, G., Stärz, M., and Lohmann, G.: Transient variability of the Miocene Antarctic ice sheet smaller than equilibrium differences, Geophysical Research Letters, 46, 4288–4298, https://doi.org/10.1029/2019GL082163, 2019.

Stap, L. B., Knorr, G., and Lohmann, G.: Anti-phased Miocene ice volume and $CO_2$ Changes by transient Antarctic ice sheet variability, Paleoceanography and Paleoclimatology, 35, e2020PA003 971, https://doi.org/10.1029/2020PA003971, 2020.

Stocchi, P., Escutia, C., Houben, A. J. P., Vermeersen, B. L. A., Bijl, P. K., Brinkhuis, H., DeConto, R. M., Galeotti, S., Passchier, S., Pollard, D., et al.: Relative sea-level rise around East Antarctica during Oligocene glaciation, Nature Geoscience, 6, 380–384, https://doi.org/10.1038/ngeo1783, 2013.

**Reply to Anonymous Referee #2**

Stap et al. perform idealized transient and steady state ice sheet model experiments, driven by GCM climate simulations, to explore Miocene ice sheet variability and hysteresis. They build on the work of Stap et al., 2019, by investigating the albedo-temperature and precipitation-ice volume feedbacks on ice growth and decay.

The strength of this work is the explicit exploration of feedbacks that influence ice sheet behavior, and the use of transient ice-sheet simulations applied to a time period that has been primarily studied using equilibrium ice sheet modeling. The weaknesses of this study (acknowledged by the authors) are the highly parameterized ice/ocean interactions, and the omission of insolation variability within the climate forcing matrix. Despite these weaknesses, I believe that this work will be of interest to the community, given some substantial clarifications described below. I suggest that the authors refocus the emphasis in this manuscript on the temperature-albedo and precipitation-ice volume feedbacks primarily.

We thank the reviewer for a careful examination of our work, and we are glad they find our study of interest. The revised manuscript will be titled '**Net effect of ice-sheet-atmosphere interactions reduces simulated transient Miocene Antarctic ice sheet variability**'', reflecting a stronger focus on the ice-sheet-atmosphere feedbacks. The experiments that regard the influence of ocean forcing on our results will be presented as a sensitivity analysis rather than robust quantifications of the thresholds for ice shelf formation and stability. Below, we respond to the reviewer and indicate how we will change the manuscript along the lines of their comments.

(1) Lack of marine ice sheets: The ice sheets simulated here are almost exclusively terrestrial. Even under the lowest $CO_2$ (280 ppm), the small WAIS seems to be grounded primarily above sea level (Fig. 1b); only with the Wilson topographies can this model setup produce marine-based ice. Therefore, the hysteresis curves presented here do not reflect marine ice dynamics. This is an understandable limitation of the study but given that geologic records show marine ice advance out onto the continental shelf during the Miocene (most recently, Pérez et al., 2021), the authors should clarify that this work cannot fully represent ice sheet volume variability through the Miocene.

Based on my understanding of their model setup, the ability of the modeled ice sheet to expand into the marine realm is primarily dependent on Tw which was linearly scaled based on $CO_2$. Could marine ice sheet advance be simulated with different choices of Tw? In the modern validation run, the authors produce marine grounded ice (because they replicate a modern ice-sheet configuration; L170, not shown). Were the Tw and basal melt rates used in this modern simulation generated using the $CO_2$ scaling, as in the Miocene runs? In other words, is the lack of marine ice in the Miocene due to the heavily parameterized basal melt / ocean temperature scheme?

The authors briefly mention this small-WAIS (no significant marine ice sheet) issue in the first paragraph of page 11, suggesting that the lack of any large WAIS growth may be hampered by not enough buildup of ice over WAIS due to climate forcing ("in the cold case simulation

1fumebi there is only a small WAIS present in the forcing climate simulation"; L333). 1fumebi is characterized by a 'full ice sheet' (L136) so I assume this is a typo, but I don't know what was meant instead. I would like the authors to elaborate on this and provide a satisfactory hypothesis (hypotheses) to explain why the 280ppm REF steady-state ice sheet under a cold orbit doesn't seem to produce marine ice advance as suggested by the geologic record and previous modeling studies (Gasson et al., 2016, Halberstadt et al., 2021, who simulate large marine-based WAIS with similar climate forcing and a steady-state ice sheet model approach). This simulation was used to initiate many of the steady state and transient simulations performed in this study, so has cascading impacts on the results here.

Indeed, our study focusses on the variability of a generally small AIS, which is located primarily on the East side of the continent, while the WAIS size remains relatively limited. This may partly be due to the BMB parameterisation, but the fact that ice shelves do grow in our REF simulations - which were run with the same $CO_2$ scaling functions for the BMB - and the simulations using Wilson topographies, suggests that the SMB is also decisive in this respect. It is therefore very important to note that our 'cold' GENESIS climate (1fumebi) was indeed generated with a relatively small WAIS; this was not a typo. We called it a 'full' ice sheet because this was how this simulation was described in Burls et al. (2021), but we agree that this is not an entirely accurate description to use in the current study. In the revised manuscript, we therefore refer to it as having a large/substantial East Antarctic ice sheet. We will also include a figure showing the ice extent used in the GENESIS simulation. Furthermore, the limitation this poses on the maximum size of the AIS in our simulations will be emphasized more strongly. Finally, the implication of our results will be discussed more clearly and extensively, also in comparison the earlier model studies of Gasson et al. (2016) and Halberstadt et al. (2021), as well as data studies among which the recent publication of Marschalek et al. (2021).

Changes: page 6 lines 173-174, page 16 lines 485-491, page 14 lines 417-419, and Figure S2.

(2) Ice shelf results: Given the heavily parameterized ocean melt scheme, I imagine that the CO2 thresholds of ice shelf formation are dependent on ocean temperature Tw which is linearly scaled with CO2. A more meaningful way to discuss these results could be to simply report the ocean temperature Tw at which ice shelves begin to form, instead of CO2. Given this dependence on the Tw scaling, I think these results should be interpreted lightly.

In the revised manuscript, we will mention the values of $T_w$ pertaining to the formation and decay of ice shelves in our REF simulations. Furthermore, the results regarding ice shelves (BMB LGM, BMB no_shelves) will be presented as essential sensitivity tests, aimed at quantifying the uncertain effect of ocean forcing on our results, rather than as robust findings of our study.

Changes: pages 3-4 lines 87-93, and page 14 lines 433-438.

Also, I am a bit confused about the interpretation presented here regarding the impact of ice shelves, for example, the CO2 thresholds above and below which ice shelves are reported to be influential (L275-276, L319-321). Looking at Fig. 8 (no-ice-shelf-melt vs. LGM ice-shelf-melt experiments), it seems to me like the identified thresholds when ice shelves are affecting

grounded ice (360 ppm and 728 ppm) are quite similar for both BMB no_shelves and BMB_LGM experiments (as well as in other 400kyr experiment that is shown; REF, Fig. 2b), so I don't quite understand the attribution to ice shelves.

Simulations BMB LGM and BMB no_shelves ease and impede the formation of ice shelves respectively. Since all other settings are kept the same, the difference between these simulations yields a quantification of the maximum influence of ocean forcing on our results. Up until the threshold for formation of ice shelves, there is no difference because the grounded ice is not in contact with the ocean. After that, the results start to diverge, reflecting the influence of the ocean forcing. The purpose of the BMB simulations, and their implications, will be explained in more detail in the revised manuscript.

*Changes: page 3-4 lines 87-93, and page 14 lines 433-443.*

The observation that ice volume variability increases when ice shelves form is straightforward and intuitive, since the role of ice shelves in buttressing grounded ice is well known. I'm not sure I see an overly significant change in "hysteresis in the CO2-V relation" (L277), though. If hysteresis is defined as the difference in CO2 initiating the onset of major ice growth and decay, there is a small difference between the solid orange and dashed green lines in the 400kyr transient runs (Fig. 8b) but very little difference in the shape of the operating curves in the 40kyr runs (Fig 8d); if hysteresis is defined as the area encompassed by the ascending and descending curve, the green dotted line does span slightly more area than the solid orange in Fig 8d but not 8b.

We indeed define hysteresis as the difference between the ascending and descending curve. We agree that in the 40-kyr runs the difference in hysteresis between the results of BMB LGM and BMB no_shelves is relatively small. However, since the grounded ice volume is consistently larger in the decay phase when ice shelves are allowed to grow, we still think we should make note of it in the manuscript.

*Changes: page 8 lines 437-438, and page 12 lines 359-360.*

(3) Elaborate in the Discussion: In my mind, the strength of this work lies in exploring the transient evolution of Miocene ice sheets and specifically investigating the impact of the albedo-temperature and precipitation-ice volume feedbacks. Therefore, it would be nice to see more analysis of transient ice sheet behavior and the impact of the two feedbacks in the Discussion section. For example, the 400 kyr cycles produce much larger ice sheets than the 40kyr cycles, suggesting that prolonged low CO2 is necessary to produce a large ice sheet. How does that relate to the geologic record of ice sheet dynamics in the Miocene? I understand that Stap et al., 2019 focused on this, but I think some discussion of how these results fit into the context of the geologic record could be summarized, and better yet, elaborated upon in the Discussion section. Given that the main contribution of this work is related to the two competing feedbacks, is there more to discuss about how these feedbacks impact hysteresis on different timescales?

Reflected by the new title, the primary focus of the revised manuscript will be the effect of ice-sheet-atmosphere interactions on transient Miocene AIS variability.

We will extend the discussion of these results with the effects on different timescales. On all the timescales studied, transient AIS variability is suppressed by the ice-sheet-atmosphere feedbacks. This is partly due to the smaller equilibrium ice volumes at low $CO_2$ levels, and partly to decreased growth rates of the transient ice volume relative to the change of equilibrium ice volume. In short, the ice-sheet-atmosphere feedbacks cause a slower build-up of the AIS to smaller peak ice volumes. We will explain in the manuscript that this implies a smaller contribution of AIS changes to Miocene benthic $\delta^{18}O$ fluctuations. However, because the actual strength of the impact will depend on the isotopic composition of Antarctic snow as well, a comprehensive quantification of these contributions is left to future efforts towards more realistic transient simulations.

All in all, we need very large $CO_2$ variations on relatively short orbital timescales (40 kyr) to obtain substantial East Antarctic ice sheet variability. This could point to a relatively stable EAIS, and a consequent reduced contribution to benthic $\delta^{18}O$ fluctuations during the Miocene. Alternatively, the WAIS could play a more major role in establishing ice-sheet variability. This will be explained and put in perspective of earlier model and data studies, in the final paragraph of the revised discussion section.

*Changes: title, page 13 lines 389-395, page 16 lines 482-491.*

*Figures*

Each figure caption should fully explain the elements of the figure or reference another figure caption where that information can be found. For example – the teal-colored areas in Figs 1, 9 are ice shelves, correct? For example, in all of the figs after Fig. 2, it would be helpful to state in the caption that the ascending branch is blue and descending branch is red. Arrows would be helpful for all figures not just Fig. 2. Also, for the equilibrium runs, I suggest adding the ice volume/CO2 points for each discrete steady state simulation.

We will aim to improve the readability of the hysteresis figures in the following ways:
- The equilibrium results are highlighted by diamonds.
- The evolution of $CO_2$ is indicated by pink and purple instead of red and blue lines in the a- and c-panels, so that it is clearly distinguishable from the equilibrium results in the b- and d-panels.
- Arrows are added to the transient results in all b- and d-panels, to indicate the progression direction.
- Legends are included to indicate which experiments are displayed.
- All the different symbols and lines are described in the figure captions.

The last item of this list also holds for the maps, where the teal - or cyan, to be precise - areas indeed indicate ice shelf extent.

*Changes:  all figures.*

Fig 9b: I wonder why there doesn't seem to be grounded or floating ice in the Ross Sea Embayment. Surely there is ice sourced from EAIS that should be able to grow into the Ross Sea with the LGM basal melt scheme?

In the BMB LGM experiments, the basal melt rate does not drop below 0 m/yr in the Ross Embayment. Therefore, the lack of ice in this region must be due to insufficient influx from the grounded West- and East-AIS regions, in combination with the (negative) SMB.

*Specific comments*

These comments will be addressed in the revised manuscript, as indicated below.

Title: When I see the phrase "influence of … solid earth on …ice sheet variability", I think of glacial isostatic adjustment and solid Earth feedbacks. I suggest replacing 'solid earth' with 'topography' in the title and elsewhere in the manuscript.

The title will be changed to 'Net effect of ice-sheet-atmosphere interactions reduces simulated transient Miocene Antarctic ice sheet variability'. Throughout the rest of the revised manuscript, references to 'solid earth' are replaced with 'bedrock topography'.

*Changes: title.*

L76: It would be helpful here (or elsewhere; L51?) to provide more information about how to read and interpret the hysteresis plots that make up the majority of the figures. For example, what exactly is meant graphically by "increased/decreased hysteresis"? Is it the total area between the ascending/descending (blue and red) curves? Or perhaps the CO2 difference between growth and collapse of the ice sheet?

We will guide the reader on how to read the hysteresis figures, upon their first occurrence when the REF experiments are described (new Fig. 1). This includes giving our general definition of hysteresis as the difference between the ascending (blue) and descending (red) branches. Mind, though, that in principle a larger difference between the thresholds for ice sheet growth and collapse also implies an increased area between the ascending and descending branches.

*Changes: page 8 lines 235-239.*

L52: Consider citing Pollard & DeConto 2005 (Hysteresis in Cenozoic Antarctic ice-sheet variations) as a seminal paper plotting hysteresis as CO2 vs ice volume.

This study will be cited in the results section, where the hysteresis curve is first discussed.

*Changes: page 8 line 237.*

L96: What are 'exposed' vs 'deep' shelf environments that the $M_{expo}$ and $M_{deep}$ melt rates are respectively applied to?

This is determined by the weighing factors $z_{deep}$ (function of water depth) and $z_{expo}$ (function of the widest subtended angle to the open ocean and the shortest linear distance to the open ocean). These factors will be introduced using numbered equations in the revised manuscript.

*Changes:  pages 4-5 lines 113-118.*

L101: Ocean temperature Tw scales linearly from -1.7 to 2 degrees C based on CO2; how was this relationship established? What assumptions does this relationship rest upon? I would imagine that this choice greatly impacts the results presented here.

Similar to Gasson et al. (2016) and Halberstadt et al. (2021), we face the limitation that GENESIS has a slab-ocean component. Realistic water temperatures can therefore not be taken from the GCM results. Instead, we use an ad hoc global ocean temperature parametrisation, adapted from the ice-sheet model ANICE (De Boer et al., 2013), the predecessor of IMAU-ICE. The experiments BMB no_shelves (severe basal melt rates) and BMB LGM (very mild basal melt rates) are performed to quantify the maximum impact of the ocean forcing on our results. They are presented as sensitivity tests in the revised manuscript.

*Changes: page 5 lines 122-124.*

L170: What ocean temperature and basal melt values were used in the modern steady state simulation, in order to match the modern ice sheet configuration? Were they based on the CO2 scaling of Tw and M? If so, that lends much more confidence to the Miocene results using that scaled approach.

Yes, they use the same $CO_2$-scaling. The modern steady-state simulations are forced with the 280-ppm $CO_2$ settings: $M_{expo}$ = 3 m/yr, $M_{deep}$ = 5 m/yr, and $T_w$ = -1.7 $^0C$.

*Changes: page 8 lines 118-119.*

L197-198: I don't understand this sentence – what are quantitative vs qualitative CO2 levels?

We mean that the ablation factor only translates the ice volumes along the $CO_2$-axis but does not change the overall shape of the $CO_2$-ice volume relation (qualitatively). Hence, the result of any change in the ablation factor, can in principle be offset by uniformly changing the forcing $CO_2$-levels.

*Changes: page 9 lines 268-276.*

L215 / Table 1: The naming convention for these experiments led to some initial confusion on my part, because the experiment FEEDB is actually removing feedbacks rather than adding them, and the REF experiment is the one that incorporates the feedbacks – so the FEEDB experiments might be better named 'NOFEEDB' or something of that sort. Similarly, the wording in this paragraph would more intuitively (to me) highlight the results by presenting the impact of adding the feedbacks rather than removing them, e.g., as in L222 ("Stated the other way around, ice-sheet-atmosphere interactions decrease the amplitude of AIS

variability"). I recommend this wording convention throughout the entire paragraph and manuscript when discussing the FEEDB (NOFEEDB) experiments.

*We agree with the reviewer on both matters. In the revised manuscript, the experiments in which the ice-sheet-atmosphere interactions are excluded, are more aptly described as using an* index method *rather than glacial index method, and are named NOFEEDB, NOFEEDB-T, and NOFEEDB-P. Furthermore, we will use the wording convention suggested by the reviewer, i.e. presenting the impact of adding, rather than removing, feedbacks.*

*Changes: page 10 lines 293-297.*

L291-295: The narrower $CO_2$ range between inception of EAIS and the marine-based WAIS in this work compared to other studies (e.g., Halberstadt et al., 2021 (and Gasson et al., 2016) seems more attributable to the different ocean melt scheme, rather than treatment of precipitation and ablation. With that said, the different precipitation and ablation schemes in previous studies probably explain the larger ice sheets they reconstruct at higher CO2 compared to this work.

*An extensive discussion of the similarities and differences between our results and those of Gasson et al. (2016) and Halberstadt et al. (2021) will be included in the final paragraph of the discussion section in the revised manuscript. A comparison to the recently published data study of Marschalek et al. (2021) will also be made there.*

*Changes: pages 15-16 lines 473-491.*

L321 "This transition from a land-based to a marine ice sheet at CO2 levels around 400 ppm is in general agreement with other model results" - I don't see evidence for a marine ice sheet in these simulations (BMB no_shelves and BMB LGM, Fig. 8, Fig. 9). In Fig 9, the 392 ppm ice sheet does not have a full WAIS, and ice volumes at 280 ppm CO2 are similar to (Fig 8b) or less than (Fig 8d) ice volume at 392 ppm (i.e., no marine ice sheet).

*We agree that we do not simulate a full modern-day-like WAIS. Nevertheless, small ice shelves are formed at 504 ppm, as indicated by the cyan-coloured areas in Fig. 9 (which will be Fig. 7 in the revised manuscript). At 392 ppm, small ice shelves fringe the entire continent. From this point onward, the ocean forcing starts to significantly affect the evolution of grounded ice volume both in the equilibrium and transient simulations. This is visible in Fig. 8 (which will be Fig. 6 in the revised manuscript) as a divergence between the results of BMB no_shelves and BMB LGM.*

*Changes: page 14 lines 431-443.*

L380: The increasing sensitivity of the AIS to a subsiding bed has been recently explored in depth (e.g., Colleoni et al., 2018; Paxman et al., 2019, 2020). The experiments presented here and corresponding discussion (L306 onwards) are interesting and relevant, but does not seem to me to produce a novel conclusion given that marine ice advance is mostly absent in these simulations and ice sheet response to a subsiding bed (and therefore increasing ice-ocean interactions) is heavily parameterized. This paragraph could be moved to the Discussion.

In the discussion section, we compare our results to the work of Colleoni et al. (2018) and Paxman et al. (2020). Our steady-state results indeed concur with their earlier findings. However, we additionally perform transient simulations. We find that the subsidence of land below sea level during the early- and mid-Miocene reduces transient 40-kyr AIS variability by 10% in amplitude. Although maybe not groundbreaking, this is to our knowledge a novel result and therefore worth mentioning in the conclusion section as well as in the abstract.

*Changes: pages 16-17 lines 512-514.*

L385: Likely tectonic evolution as well as glacial erosion

This sentence will be removed from the manuscript.

*Changes: page 17 line 514 (deletion).*

**REFERENCES:**

Colleoni, F., De Santis, L., Montoli, E., Olivo, E., Sorlien, C. C., Bart, P. J., Gasson, E. G. W., Bergamasco, A., Sauli, C., Wardell, N., et al.: Past continental shelf evolution increased Antarctic ice sheet sensitivity to climatic conditions, Scientific Reports, 8, 1–12, https://doi.org/10.1038/s41598-018-29718-7, 2018.

De Boer, B., Van de Wal, R. S. W., Lourens, L. J., Bintanja, R., and Reerink, T. J.: A continuous simulation of global ice volume over the past 1 million years with 3-D ice-sheet models, Climate Dynamics, 41, 1365–1384, https://doi.org/10.1007/s00382-012-1562-2, 2013.

Gasson, E., DeConto, R. M., Pollard, D., and Levy, R. H.: Dynamic Antarctic ice sheet during the early to mid-Miocene, Proceedings of the National Academy of Sciences, 113, 3459–3464, https://doi.org/10.1073/pnas.1516130113, 2016.

Halberstadt, A. R. W., Chorley, H., Levy, R. H., Naish, T., DeConto, R. M., Gasson, E., and Kowalewski, D. E.: CO2 and tectonic controls on Antarctic climate and ice-sheet evolution in the mid-Miocene, Earth and Planetary Science Letters, 564, 116908, https://doi.org/10.1016/j.epsl.2021.116908, 2021.

Marschalek, J. W., Zurli, L., Talarico, F., van de Flierdt, T., Vermeesch, P., Carter, A., Beny, F., Bout-Roumazeilles, V., Sangiorgi, F., Hemming, S. R., et al.: A large West Antarctic Ice Sheet explains early Neogene sea-level amplitude, Nature, 600, 450–455, https://doi.org/10.1038/s41586-021-04148-0, 2021.

Paxman, G. J. G., Gasson, E. G. W., Jamieson, S. S. R., Bentley, M. J., and Ferraccioli, F.: Long-Term Increase in Antarctic Ice Sheet Vulnerability Driven by Bed Topography Evolution, Geophysical Research Letters, 47, e2020GL090003, https://doi.org/10.1029/2020GL090003, 2020.

---

## Author Response (AR2)

**Comments to the authors:**

To the authors,

Thank you very much for submitting your revised manuscript. You have thoroughly addressed all reviewer comments and the manuscript looks like it is in very good shape. I have only found some minor grammatical changes (listed below) that should be considered before final publication.

Best regards,
Alex

Minor comments:

L14: reduces ice volume variability => also reduces ice-volume variability

L18: (paleo)climate => paleoclimate

L87: threedimensional => three-dimensional

L149: dynamical sea ice => dynamic sea ice

L251: "This also results..." <= Consider replacing "This" with the explicit name of what you are referring to here. It is not completely clear to me.

L260: this experiment => the latter experiment

L273: floatation => flotation

L385: floatation => flotation [Check for other instances...]

**Reply by the authors:**

We have implemented all the suggestions given by the Editor. This includes the change of all instances of the word 'floatation' to 'flotation' in both the manuscript and the supplement. We thank the Editor very much for handling our submission.